# Changes in South American surface ozone trends: exploring the influences of precursors and extreme events

**Rodrigo J. Seguel**[1,2], **Lucas Castillo**[1,2], **Charlie Opazo**[1,2], **Néstor Y. Rojas**[3], **Thiago Nogueira**[4], **María Cazorla**[5], **Mario Gavidia-Calderón**[6], **Laura Gallardo**[1,2], **René Garreaud**[1,2], **Tomás Carrasco-Escaff**[1], **and Yasin Elshorbany**[7]

[1]Center for Climate and Resilience Research (CR)[2], Santiago, Chile TS1
[2]Department of Geophysics, Faculty of Physical and Mathematical Sciences,
University of Chile, Santiago, Chile
[3]Department of Chemical and Environmental Engineering,
Universidad Nacional de Colombia, Bogotá, Colombia
[4]Departamento de Saúde Ambiental, Faculdade de Saúde Pública, Universidade de São Paulo, São Paulo, Brazil
[5]Instituto de Investigaciones Atmosféricas, Universidad San Francisco de Quito USFQ, Quito, Ecuador
[6]Departamento de Ciências Atmosféricas, Instituto de Astronomia, Geofísica e Ciências Atmosféricas,
Universidade de São Paulo, São Paulo, Brazil
[7] CE1 University of South Florida, St. Petersburg, Florida, USA TS2

**Correspondence:** Rodrigo J. Seguel (rodrigoseguel@uchile.cl)

**Abstract.** TS3 In this study, trends of 21st-century ground-level ozone and ozone precursors were examined across South America, a less-studied region where trend estimates have rarely been comprehensively addressed. Therefore, we provided an updated regional analysis based on validated surface observations. We tested the hypothesis that the recent increasing ozone trends, mostly in urban environments, resulted from intense wildfires driven by extreme meteorological events impacting cities where preexisting volatile organic compound (VOC)-limited regimes dominate. We applied the quantile regression method based on monthly anomalies to estimate trends, quantify their uncertainties and detect trend change points. Additionally, the maximum daily 8 h average (MDA8) and peak-season metrics were used to assess short- and long-term exposure levels, respectively, for the present day (2017–2021). Our results showed lower levels in tropical cities (Bogotá and Quito), varying between 39 and 43 nmol mol$^{-1}$ for short-term exposure and between 26 and 27 nmol mol$^{-1}$ for long-term exposure. In contrast, ozone mixing ratios were higher in extratropical cities (Santiago and São Paulo), with a short-term exposure level of 61 nmol mol$^{-1}$ and long-term exposure levels varying between 40 and 41 nmol mol$^{-1}$. Santiago (since 2017) and São Paulo (since 2008) exhibited positive trends of 0.6 and 0.3 nmol mol$^{-1}$ yr$^{-1}$, respectively, with very high certainty. We attributed these upward trends, or no evidence of variation, such as in Bogotá and Quito, to a well-established VOC-limited regime. However, we attributed the greater increase in the extreme percentile trends ($\geq$ 90th) to heat waves and, in the case of southwestern South America, to wildfires associated with extreme meteorological events.

# 1 Introduction

The global tropospheric ozone ($O_3$) burden has increased by 45 % ($109 \pm 25$ Tg) with medium confidence from 1850 to the present day due to anthropogenic precursor emissions (Szopa et al., 2021). Additionally, surface ozone has increased by 32 %–71 % with large uncertainty in rural air across the Northern Hemisphere relative to historical observations (1896–1975) (Tarasick et al., 2019). Since the mid-1990s, free tropospheric ozone has increased with high confidence by 1–4 nmol $mol^{-1}$ per decade in most regions across the northern mid-latitudes and 1–5 nmol $mol^{-1}$ per decade within the tropics (Guleb et al., 2021 TS4). In contrast, the identification of ozone trends in the Southern Hemisphere, including South America, is precluded due to the limited coverage by ground-level monitoring stations, while observations of tropospheric column ozone since the mid-1990s indicate medium confidence increases of less than 1 nmol $mol^{-1}$ per decade at southern mid-latitudes (Gulev et al., 2021 TS5; Cooper at al., 2020)

South America (10° N to 55° S) encompasses tropical, subtropical and mid-latitude climates, in addition to high-altitude conditions within the Andes mountain range (Garreaud et al., 2009). This continent also hosts essential ecosystems for global water and carbon cycles, including tropical and subtropical wetlands, Andean glaciers, the Amazon rainforest and peatlands in Patagonia (Gumbricht et al., 2017; Hoyos-Santillan et al., 2019; Heinrich et al., 2021; Lapere et al., 2023; Molina et al., 2015). In contrast, 85 % of the continent's population, estimated at more than 430 million people in 2022, resides in urban areas (Population Reference Bureau, 2024). In addition to geographical and climatological contexts, cities vary in terms of public transportation, industry, governance, regulation and the degree of clean energy penetration (Cazorla et al., 2022).

In South American countries, the enactment of air quality standards started in the 1990s and has progressively increased. Except for a few cities, air quality monitoring in the region provides records spanning no longer than 1 or 2 decades, contrasting with the spatial coverage and long-term records in other regions, such as North America and Europe (Schultz et al., 2017). Most countries set ozone standards based upon an 8 h average, ranging from 51 to 71 nmol $mol^{-1}$ (Lyu et al., 2023). However, ozone regulation does not guarantee compliance, as observed in cities such as Santiago and São Paulo, which have been designated nonattainment areas for decades (Seguel et al., 2020; Andrade et al., 2017). Furthermore, photochemical pollution has generally received less attention than particle pollution despite the increasing role of photochemistry in secondary aerosol formation (Andrade et al., 2017; Menares et al., 2020).

In most urban areas with adequate monitoring coverage allowing characterization of the temporal and spatial variabilities in ground-level ozone, a chemical regime of ozone formation limited by volatile organic compounds (VOCs) has been found in previous work (Seguel et al., 2020; Silva et al., 2018; Silva Júnior et al., 2009; Elshorbany et al., 2009). This chemical regime was also observed during COVID-19 pandemic lockdowns, when several cities experienced increased ambient ozone mixing ratios (e.g., Bogotá, Quito, Santiago, São Paulo and Lima) due to a decrease in nitric oxide (NO) emitted by motorized transportation vehicles (Seguel et al., 2022; Sokhi et al., 2021; Cazorla et al., 2021).

Monitoring has also been implemented through the Global Atmospheric Watch (GAW) program of the World Meteorological Organization (WMO) at remote locations, which allows the study of changes in the chemical composition at high-altitude sites (e.g., Tololo and Chacaltaya at 2200 and 5500 m a.s.l., respectively) and at pristine locations such as Patagonia (Ushuaia) and Galapagos (Anet et al., 2017; Cazorla and Herrera, 2020). Background monitoring stations are essential for interpreting the feedback between the atmospheric composition and the intensification of extreme weather events as well as for studying changes in transport patterns that could favor stratosphere–troposphere exchange (STE) in the extratropics (Lu et al., 2019; Cooper et al., 2020).

In South America, temperature trends have increased, except along the west coast (Perú and Chile) (Gu and Adler, 2023; Falvey and Garreaud, 2009). The most significant warming has occurred in the tropics and central part of the continent. In contrast, opposite precipitation trends have been observed in different regions of South America. Notable drying has been found in central Chile and the southern part of the Amazon basin (Gu and Adler, 2023). Under this scenario, the combination of megadroughts (Garreaud et al., 2020), warmer summers and more frequent heat waves (Jacques-Coper et al., 2021; González-Reyes et al., 2023) provides favorable conditions for the onset of large-scale fires and subsequent emission of ozone precursors (Feron et al., 2023). However, the distribution of these air masses is complex and must be examined further, given the complexity of the topography and disruption of airflow by the Andes Mountain range. Moreover, the southern portions of South America are subject to long-range transport of ozone and its precursors, as demonstrated by the arrival of plumes derived from fires in 2019 and large amounts of biomass burning (Kloss et al., 2021; Daskalakis et al., 2022).

In this study, we examined the distribution and trends in ozone and its precursors in cities and background locations in South America and analyzed trend change points. We propose that the precursor ratio (nitrogen oxides to VOC) largely determines the observed ozone trends in South American urban environments, while short-lived but increasingly recurrent extreme weather events (high temperatures, low relative humidity levels and moderate to high winds) may also impact ozone trends. By providing this updated regional analysis of the distributions and trends in ozone and its precursors, we contribute to a better understanding of trend changes within

regional, hemispheric and global contexts and identify information gaps to be resolved by the scientific community.

## 2 Methodology

Time series of surface-level ozone, nitrogen oxides ($NO_x$), carbon monoxide (CO) and meteorological variable data were obtained from governmental agencies in Bogotá, Colombia; Chile; Ecuador; and the State of São Paulo, Brazil. The ozone measurement principle is based on a UV absorption technique, $NO_x$ on chemiluminescence, where $NO_2$ is converted to NO by a molybdenum converter heated before detection, and CO is measured using an infrared absorption technique. These agencies follow their own data policies, including quality assurance–quality control procedures and traceability. Nonetheless, the authors assessed the datasets by applying further checks: (1) we evaluated the completeness by considering time series with at least 75 % of daily, monthly and annual data and 3 years of consecutive measurements; (2) for each time series, we assessed (expert judgment) the remaining data gaps and the instrument response after those gaps; (3) we check the integrity of the time series now based on monthly anomalies; and finally (4) precursors' time series that do not measure ozone at the monitoring site were omitted. As a result of this process, we produced a homogenized dataset, which was utilized in this research and submitted to the database of the Tropospheric Ozone Assessment Report, phase II (TOAR-II).

The 21st-century trend analyses described in Sect. 2.2 were performed individually for 74 air quality monitoring stations and 3 stations of the GAW program. Only 7 monitoring stations located in medium-sized cities (between $\sim 50\,000$ and $\sim 300\,000$ inhabitants), mainly in Chile, were included in this study due to the completeness of the data and lack of long-term time series. A similar situation occurred for some WMO stations and high-altitude sites, which did not pass the 75 % filter of valid data.

The time series retrieved from stations in the main cities were aggregated to provide the trend and uncertainty in the monitoring network of each city. Such aggregation and subdivision operations were performed according to local expert judgment, thereby accounting for representativity (human health, baseline, industrial influence), altitude, topography and precursor sources.

We used CO data in the absence of systematic measurements of VOCs in South America to explore the ozone chemical formation regime. Since carbon monoxide is typically co-emitted during fuel combustion and transport in urban environments, this pollutant can be used as a surrogate for anthropogenic VOCs. Our trend analysis did not explicitly provide the CO-to-$NO_x$ ratio because the available time series exhibited different lengths and gaps. Therefore, we obtained CO, NO (nitrogen dioxide, $NO_2$) and $NO_x$ trends individually to infer the ozone formation regime, as well as their respective change points.

**Table 1.** Trend reliability scale according to TOAR-II recommendations (Chang et al., 2023).

| $p$ value | SNR value | Term |
|---|---|---|
| $p \leq 0.01$ | $SNR \geq 3$ | Very high certainty |
| $0.05 \geq p > 0.01$ | $2 \leq SNR < 3$ | High certainty |
| $0.10 \geq p > 0.05$ | $1.65 \leq SNR < 2$ | Medium certainty |
| $0.33 \geq p > 0.10$ | $1 \leq SNR < 1.65$ | Low certainty |
| $p > 0.33$ | $SNR < 1$ | Very low certainty or no evidence |

### 2.1 Short- and long-term ozone metric evaluation

Present-day short- and long-term ozone exposure levels were assessed for all stations available following the TOAR-II recommendation for timescales, i.e., averaged values across 2017–2021, to facilitate intercomparisons with other studies. To evaluate short-term exposure, we utilized the 99th percentile of the annual distribution of the maximum daily 8 h average (MDA8) mixing ratio. For long-term exposure, we used the peak-season value proposed by the World Health Organization (WHO, 2021), defined as the average MDA8 mixing ratio calculated for the 6 consecutive months of the year with the highest 6-month running-average ozone mixing ratio. To identify sites at exposure risk, we used the MDA8 and peak-season guidelines, at 51 and 31 nmol mol$^{-1}$, respectively (WHO, 2021).

### 2.2 Trend analysis

We calculated daily and monthly means based on hourly data, ensuring 75 % valid data each day and month. An anomaly was determined by calculating the difference between harmonic functions (6 and 12 months) and the observed monthly value.

Subsequently, we applied the quantile regression (QR) method to conduct trend analysis based on monthly anomalies following TOAR-II recommendations (Chang et al., 2023). We applied a moving-block bootstrap algorithm to account for the autocorrelation and calculated standard errors of the trends. Trend uncertainty was expressed using a calibrated language based on the $p$ value and signal-to-noise (SNR) ratio (i.e., trend and standard error) (Table 1). The adopted scale graduation aimed to communicate the trend reliability across TOAR-II Working Groups (WGs).

We used the piecewise linear trend method to detect change points in the time series (Muggeo, 2003). A detected change point was considered valid when the magnitude of the SNR values before and after the change point was maximized ($\geq 2$). We imposed a minimum period of 4 years after the occurrence of a change point to avoid detection at the

extremes of the time series. The SNR used to detect change points was obtained from the piecewise linear trend (it is not the same SNR obtained from quantile regression).

We utilized the following linear regression model to calculate the trend in the entire time series:

$$y_t = \alpha + \beta t + \sum_{k}^{N} \gamma_k \max\{t - x_{c_k}, 0\} + \varepsilon, \qquad (1)$$

where $y_t$ is the monthly anomaly of the variable, $t$ is the monthly index of the observed anomaly, $\alpha$ is a constant and $\beta$ is the linear trend before the first change point $x_{c_1}$. If any exists, $x_{c_k}$ is the index of the change point; $\gamma_k$ denotes the linear trend after each change point, with $k = 1, 2, \ldots, N$; $N$ is the number of change points detected for each time series; and $\max\{t - x_{c_k}, 0\}$ is equal to 1 for $t > x_{c_k}$ and 0 for $t \le x_{c_k}$. When no change point is incorporated, the linear regression model is reduced to $y_t = \alpha + \beta t$. According to our linear regression model, the time series trend after the first change point is $\beta + \gamma_1$, and that after the second change point is $\beta + \gamma_1 + \gamma_2$. To calculate the standard error of the trend after a change point, the bootstrap algorithm is again applied over the time series segment. Finally, $\varepsilon$ denotes the error of the linear regression.

## 3 Results and discussion

Major urban agglomerations are distributed in different latitudinal bands, altitudes and regions of South America, enabling regional interpretation despite the limited monitoring sites. Quito (0.12° S, 78.5° W; 2800 m a.s.l.) and Bogotá (4.6° N, 74.1° W; 2600 m a.s.l.) are located at high altitudes in the northwestern South American region within the tropical latitudinal band (defined between 20° N and 20° S). São Paulo (23.5° S, 70.5° W; 740 m a.s.l.) is located in southeastern South America within the subtropical band (defined between 20 and 30° S), and Santiago (33.5° S, 70.5° W; 500 m a.s.l.) is located in southwestern South America at the equatorward edge of the mid-latitude band (defined between 30 and 60° S).

### 3.1 Ozone distribution overview in urban environments: MDA8 and the peak-season value

The MDA8 metric calculated for the present day (2017–2021) showed that the lowest short-term exposure levels were found in Bogotá (43 nmol mol$^{-1}$) and Quito (39 nmol mol$^{-1}$), both of which are located in the tropical band. In contrast, the highest levels of MDA8 were observed in Santiago (mid-latitude) and São Paulo (subtropics), at 61 nmol mol$^{-1}$. Figure 1 shows the MDA8 value for each available air quality monitoring station, which indicates, in some cases, the main receptor sites in these urban agglomerations. Additionally, Table 2 lists the present-day ozone exposure metrics for the aggregated stations in the subdivisions

used and includes the preceding 5 years (2012–2016) to provide a frame of comparison. In this regard, central and northern Bogotá, eastern Quito, northeastern Santiago (not shown in Table 2 (77 nmol mol$^{-1}$)) and the city of São Paulo exhibited the highest MDA8 levels. Additionally, in São Paulo, the MDA8 metric was significantly lower at present than in the preceding 5 years.

Overall, the data suggested different spatial distributions of short- and long-term exposure to ozone within the cities considered in South America, as both metrics (MDA8 and peak-season value) were substantially lower in Quito and Bogotá than at the Santiago and São Paulo stations. This finding is consistent with processes that occurred in the tropics such as higher ozone photolysis ($\lambda \le 336$ nm) promoted by the intense solar ultraviolet radiation, high humidity favoring water reaction with atomic oxygen (O($^1$D)) and the strong convection producing upward airflow, resulting in short O$_3$ lifetime (Kley et al., 1996). Other aspects related to ozone precursors will be discussed in Sect. 3.3.

Bogotá, Quito, Santiago and São Paulo are busy urban centers where urban emissions comprise a significant source of ozone precursors. In these cities, a significant fraction of ozone precursors is emitted by vehicular fleets and has decreased according to air quality control measures such as the introduction of better fuel quality, enforcement of three-way catalytic converters, stricter emission standards for new fleet vehicles and mandatory periodic technical inspection for in-use vehicles (Andrade et al., 2017; Nogueira et al., 2021; Osses et al., 2022). In addition to anthropogenic emissions, these cities exhibit particularities explaining the ozone and surface exposure levels reached, which are described below.

Ozone precursor transport changes seasonally on the Bogotá Plateau. Southeasterly winds dominate most of the day between April and September. These predominant winds are associated with relatively low ozone levels due to the presence of fewer upwind sources. From October to March, westerly winds in the afternoon hours are associated with the highest ozone levels and air quality standard exceedances (61 nmol mol$^{-1}$), especially in the northeast region of the city, which receives more photochemically processed air masses.

In Quito, ozone levels usually remain below the national air quality standard (51 nmol mol$^{-1}$), and short-term exposure levels are less than 40 nmol mol$^{-1}$ (present day), as indicated earlier. Bogotá and Quito exposure levels are consistent with ozone profile in situ measurements over the South American tropics, where the 50th (5th) percentile was found to be less than 40 nmol mol$^{-1}$ (10 nmol mol$^{-1}$) from the surface to 200 hPa (700 hPa) (Gaudel et al., 2024).

In the Santiago Basin, ozone precursor emissions in the city are typically transported toward the northeast during the afternoon hours, especially during the summer period (December–March), when a stronger valley-to-mountain breeze occurs. Santiago experiences a marked seasonal cycle that reaches a minimum ozone level in the winter months

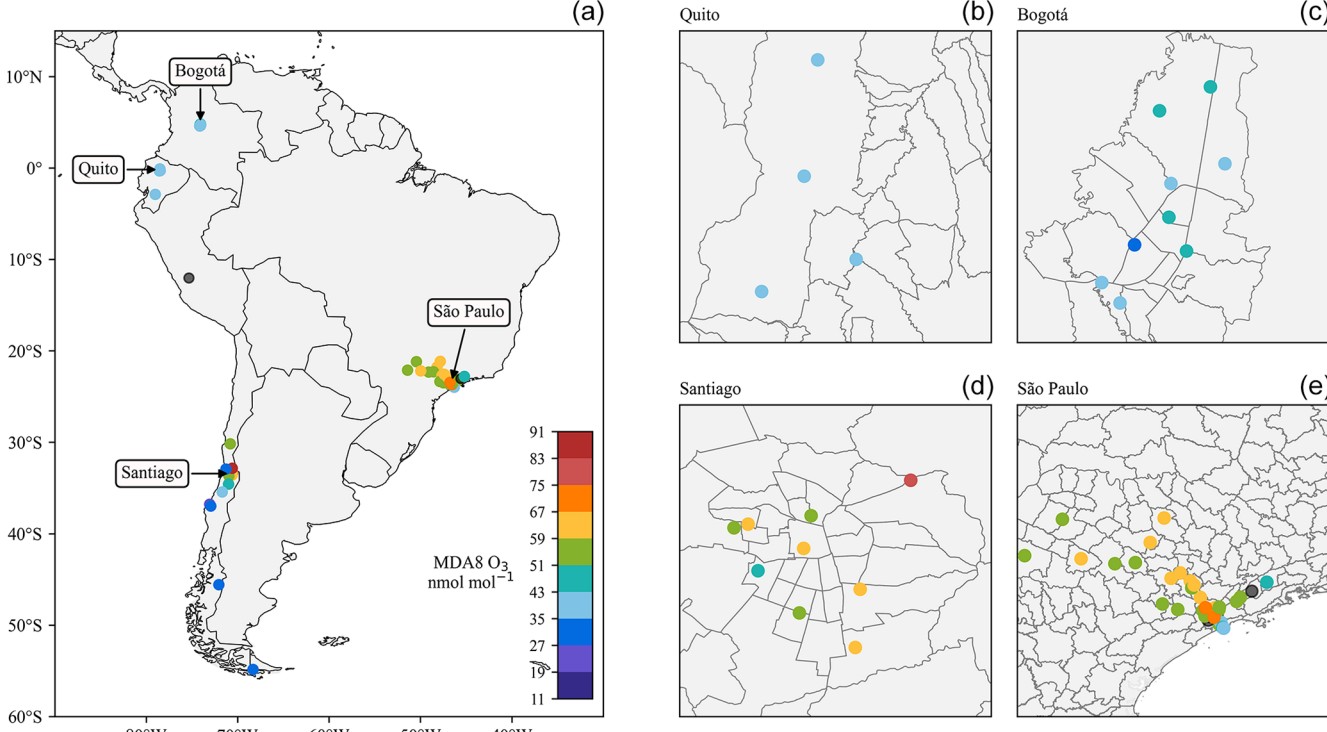

**Figure 1.** Present-day MDA8 ozone (2017–2021) in nmol mol$^{-1}$ calculated from available South American monitoring stations **(a)**. Panels **(b)**–**(e)** focus on Bogotá, Quito, Santiago and São Paulo. The grey dots denote monitoring stations that do not meet the data quality criteria.

**Table 2.** MDA8 and peak-season metrics calculated from 2012–2016 and 2017–2021 (present day) with 95 % confidence intervals and percentage changes in the urban agglomerations and subdivisions.

| Subdivisions | MDA8 (nmol mol$^{-1}$) | | | Peak season (nmol mol$^{-1}$) | | |
|---|---|---|---|---|---|---|
| | 2012–2016 | 2017–2021 | Change (%) | 2012–2016 | 2017–2021 | Change (%) |
| Bogotá | 42.2 [± 4.2] | 42.8 [± 3.6] | 1.4 | 24.9 [± 2.4] | 26.4 [± 3.1] | +5.9 |
| Northern Bogotá | 48.7 [± 1.2] | 46.2 [± 3.8] | −5.0 | 27.5 [± 2.4] | 27.6 [± 2.0] | +0.16 |
| Central Bogotá | 45.1 [± 6.1] | 46.6 [± 7.3] | 3.2 | 27.0 [± 3.2] | 29.7 [± 6.4] | +9.7 |
| Southwest Bogotá | 35.6 [± 1.4] | 38.2 [± 8.9] | 7.3 | 22.0 [± 6.3] | 25.2 [± 8.9] | +14 |
| Southeast Bogotá | 35.7 [± 4.8] | 39.1 [± 1.5] | 9.5 | 22.3 [± 6.9] | 22.2 [± 1.5] | −0.52 |
| Quito | 41.1 [± 1.7] | 38.6 [± 1.9] | −6.0 | 26.3 [± 1.2] | 26.7 [± 1.1] | +1.5 |
| Quito main urban area | 40.0 [± 2.9] | 36.9 [± 1.4] | −7.5 | 26.1 [± 1.3] | 26.3 [± 2.0] | +1.1 |
| Quito eastern valleys | 42.2 [± 2.2] | 41.5 [± 1.4] | −1.7 | 26.5 [± 3.9] | 27.4 [± 1.5] | +3.5 |
| Santiago | 61.2 [± 4.4] | 60.7 [± 5.5] | −0.83 | 39.1 [± 2.6] | 40.9 [± 2.9] | +4.5 |
| São Paulo (SPMA and city) | 76.6 [± 2.6] | 61.3 [± 2.6] | −20 | 42.8 [± 1.9] | 40.1 [± 1.7] | −6.1 |
| São Paulo Metropolitan Area (SPMA) | 77.7 [± 5.4] | 60.6 [± 3.4] | −22 | 42.3 [± 3.0] | 38.5 [± 1.4] | −8.9 |
| City of São Paulo | 76.1 [± 3.0] | 61.7 [± 3.6] | −19 | 43.0 [± 2.4] | 40.9 [± 2.3] | −4.8 |
| Coastal São Paulo | 53.1 [± 5.2] | 42.7 [± 4.9] | −20 | 29.0 [± 2.1] | 28.4 [± 2.3] | −2.1 |
| Industrial São Paulo | 64.2 [± 5.8] | 47 [± 13] | −27 | 29.3 [± 7.0] | 28.9 [± 8.0] | −1.2 |
| Inland São Paulo | 64.4 [± 5.2] | 57.9 [± 2.2] | −10 | 42.8 [± 1.7] | 42.3 [± 1.7] | −1.2 |

and a maximum in late summer, usually peaking in March when late-summer high temperatures are combined with high post-holiday vehicle activity. Large-scale subsidence conditions and thermally driven valley–mountain circulation result in the accumulation of ozone and ozone precursors aloft during summertime, which can increase surface ozone when a well-mixed boundary layer develops in the afternoon (Seguel et al., 2013; Lapere et al., 2021). Thus, the northeastern region of the city is characterized by the highest ozone standard exceedances.

In the State of São Paulo, the main source of air pollution and ozone precursors is vehicular fleets, which widely use biofuels such as ethanol, gasohol (ethanol and gasoline mixture) and biodiesel (Andrade et al., 2017). High ozone mixing ratios are typically measured at the end of winter and in spring (late August to November), when clear-sky conditions favor ozone formation (Carvalho et al., 2015). Additionally, in these months, biomass burning emissions from sugarcane burning occur in inland São Paulo, which can reach the São Paulo Metropolitan Area. As shown in Fig. 1, high ozone mixing ratios can occur in inland São Paulo, caused by the transport of ozone precursors from the São Paulo Metropolitan Area under predominant southeasterly winds and inland biomass burning emissions (Squizzato et al., 2021).

Mid-sized cities in South America, in contrast, face several challenges in terms of monitoring station availability and quality control. Notably, in Chile, out of 75 ozone monitoring stations, only 18 (9 located in Santiago) passed the quality control test established in the methodology, which is a warning in terms of the efficient use of resources.

Cuenca in Ecuador and southern Chilean mid-sized cities ($< 34°$ latitude) have low MDA8 levels. Coyhaique ($45.6°$ S, $72.1°$ W; 356 m a.s.l.), located in southern Chile, which otherwise exhibits extremely high particle levels in winter due to wood burning, attains an MDA8 level of 27.3 nmol mol$^{-1}$. In contrast, Los Andes in Chile ($32.8°$ S, $70.6°$ W; 819 m altitude) exhibits the highest MDA8 level in South America (87.5 nmol mol$^{-1}$). Los Andes has been described as a typical receptor site for air pollutants originating in the upwind Santiago Metropolitan Region (Seguel et al., 2013) and potentially from coastal areas, where high levels of VOCs have been measured in highly concentrated industrial zones (Seguel et al., 2023).

According to the long-term exposure metric, millions of inhabitants in Santiago and São Paulo are exposed to unhealthy ozone levels (Fig. 2). Again, low peak-season levels were observed in Bogotá (26 nmol mol$^{-1}$) and Quito (27 nmol mol$^{-1}$). In contrast, peak-season exposure exceeded the 31 nmol mol$^{-1}$ guideline in Santiago (41 nmol mol$^{-1}$) and in some State of São Paulo subdivisions, such as the city of São Paulo, São Paulo Metropolitan Area and inland São Paulo (Table 2). Only coastal São Paulo and the industrial zone subdivisions occurred below the peak-season guideline.

In contrast to short-term exposure, which shows signs of improvement, mainly in São Paulo, present-day long-term exposure has led to more discrete improvements in São Paulo and increases in Bogotá, Quito and Santiago. Again, Los Andes in Chile attained the highest peak-season value recorded (58 nmol mol$^{-1}$), followed by northeast Santiago (49 nmol mol$^{-1}$).

## 3.2 Ozone trends at urban and background sites

Ozone trends in the 50th percentile and reliability are shown in Fig. 3. The trend after the last detected change point (if any) is shown in Fig. 3. Most change points detected in the time series occurred after 2010 (Fig. A1 in Appendix A). In addition, Table 3 shows the extreme percentiles (5th and 95th) and the median trend in large cities. Santiago and São Paulo exhibited clear positive trends, with high or very high certainty after the change points in 5th, 50th and 90th percentiles (Table 3). The carbon monoxide trend is not clear (considering the calculated uncertainties) except for some subdivisions in São Paulo, where CO drastically declines. The Bogotá, Quito and Santiago stations showed an upward CO trend and a very low certainty (Figs. S1 and S2 in the Supplement). Nitric oxide and nitrogen dioxide showed a downward trend (with a few exceptions)(Figs. S3–S8). Both precursors are analyzed in more detail in the next section.

Three background monitoring stations are shown in Fig. 3, including two high-altitude sites (Huancayo in Peru and Tololo in Chile), which are essential for studying the impact of stratosphere–troposphere transport on ground-level ozone. Huancayo (3310 m a.s.l.; $12.04°$ S, $75.32°$ W) shows no evidence of increasing or decreasing ozone trends. The Tololo station, located in the subtropical Andean region (2220 m a.s.l.; $30.17°$ S, $70.80°$ W), is one of the two WMO stations in South America with sufficiently long and complete time series to evaluate the impacts of both STE and poleward expansion of the Hadley circulation. At Tololo, an upward ozone trend of 0.29 nmol mol$^{-1}$ yr$^{-1}$ was observed between 2006 and 2014 (Fig. 4), with a very high certainty (Table 3). Within this period (2006–2014), the higher percentiles ($> 50$th) displayed the most significant increasing trends ($> 0.38$ nmol mol$^{-1}$ yr$^{-1}$). Notably, the trend change point in May 2006 $\pm$ [November 2002, November 2009] coincides with the global methane increase after the plateau observed between 1999 and 2006 (Lan et al., 2024). These observational findings could be explored further by comparing them with outputs from regional models capable of quantifying the ozone increase associated with methane changes. On the other hand, the second change point confidence interval (2014) was very wide compared with the length of the time series, as shown in Fig. 4, and although the trend for this period was still positive (0.07 nmol mol$^{-1}$ yr$^{-1}$), the certainty was relatively low. Moreover, the trend observed after 2014 was likely impacted by ozone drops due to the COVID-19 pandemic in 2020 and possibly in 2021, as found at high-

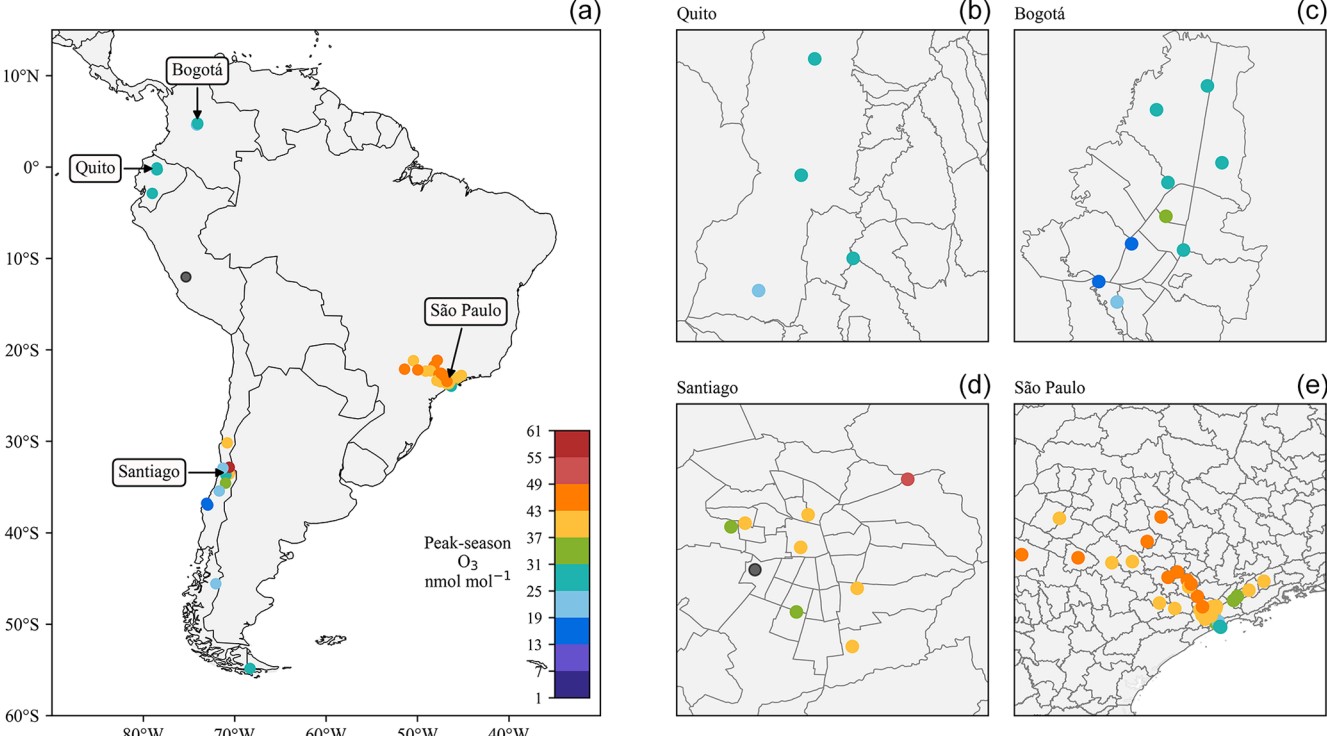

**Figure 2.** Peak-season ozone from 2017–2021 in nmol mol$^{-1}$ calculated from available South American monitoring stations (**a**). Panels (**b**)–(**e**) focus on Bogotá, Quito, Santiago and São Paulo. The grey dots denote the monitoring stations that do not meet the data quality criteria.

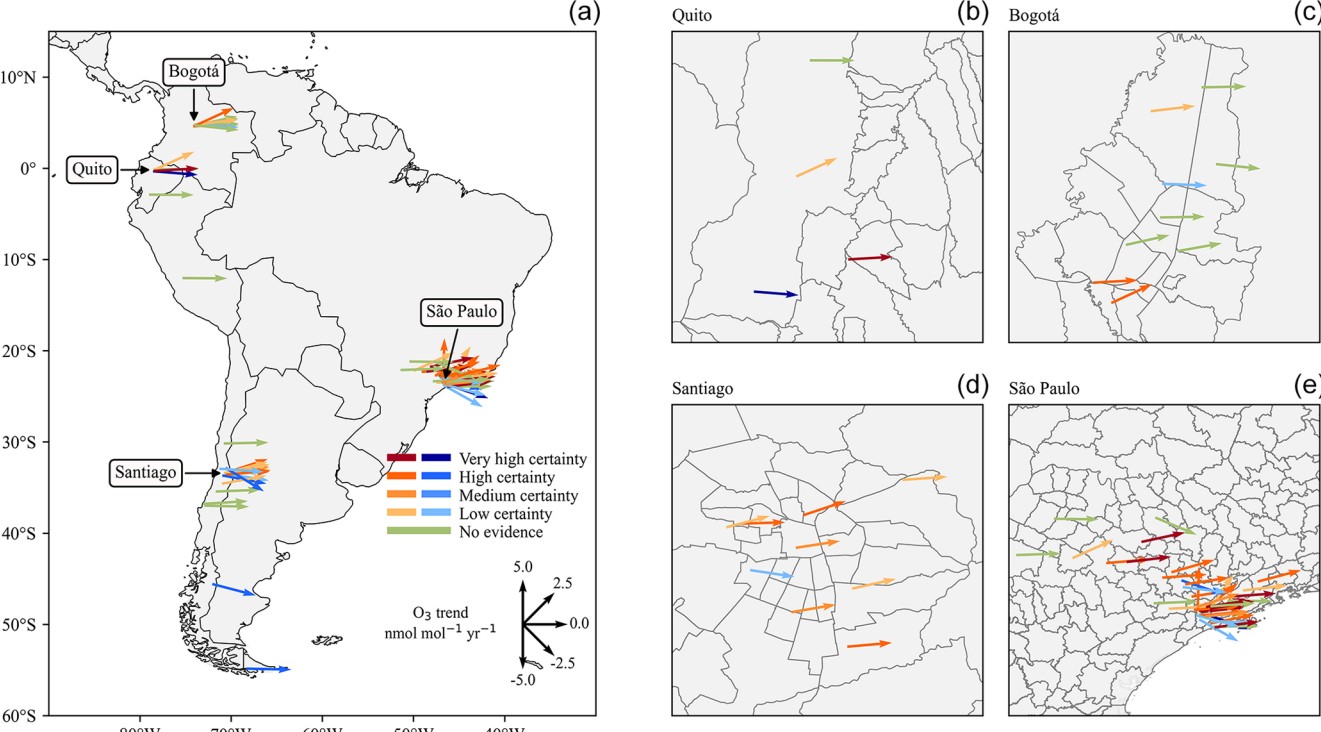

**Figure 3.** Ground-level ozone trends in nmol mol$^{-1}$ yr$^{-1}$ and reliability levels calculated from available South American monitoring stations (**a**). When one or more change points are identified, the trend starts from the latest change point detected. Panels (**b**)–(**e**) focus on Bogotá, Quito, Santiago and São Paulo.

I found some minor errors in the p-value (my mistake). I would appreciate it if you could change it with the numbers I am providing right next to the table. Note that this change does not affect the reliability column or any conclusion in the manuscript.

**Table 3.** CE2 Ozone trends and reliability estimated for large cities and Tololo station (5th, 50th (median) and 95th percentiles). The table also provides the time series period for each city, number of monitoring stations considered and year of the change point.

| Location (no. stations) | Time series length | Percentile | Change points | Piecewise dSNR | Trend (nmol mol$^{-1}$ yr$^{-1}$) | SNR | $p$ value | Reliability | |
|---|---|---|---|---|---|---|---|---|---|
| Bogotá (13) | January 2008– April 2021 | 5th | Not detected | – | 0.03 [± 0.25] | 0.21 | $8.34 \times 10^{-1}$ | Very low certainty | |
| | | 50th | Not detected | | −0.02 [± 0.19] | −0.22 | $8.26 \times 10^{-1}$ | Very low certainty | |
| | | 95th | Not detected | | −0.06 [± 0.41] | −0.30 | $7.67 \times 10^{-1}$ | Very low certainty | |
| Quito (6) | September 2005– January 2022 | 5th | Before April 2011 | 4.2 | −0.3 [± 0.33] | −1.84 | $6.67 \times 10^{-2}$ | Medium certainty | Instead 65 it is 23 |
| | | 5th | After April 2011 | | 0.05 [± 0.16] | 0.64 | $5.65 \times 10^{-1}$ | Very low certainty | |
| | | 50th | Before April 2011 | | −0.22 [± 0.21] | −2.14 | $3.37 \times 10^{-2}$ | High certainty | |
| | | 50th | After April 2011 | | 0.06 [± 0.14] | 0.90 | $4.10 \times 10^{-1}$ | Very low certainty | 3.68 |
| | | 95th | Before April 2011 | | −0.63 [± 0.75] | −1.67 | $9.57 \times 10^{-2}$ | Medium certainty | |
| | | 95th | After April 2011 | | 0.11 [± 0.33] | 0.68 | $3.22 \times 10^{-1}$ | Very low certainty | 4.96 |
| Santiago (9) | April 1997– April 2023 | 5th | Before November 2017 | 19 | −0.20 [± 0.07] | −5.53 | $6.70 \times 10^{-8}$ | Very high certainty | |
| | | 5th | After November 2017 | | 0.72 [± 0.40] | 3.65 | $4.60 \times 10^{-2}$ | Very high certainty | 5.27x10-4 |
| | | 50th | Before November 2017 | | −0.19 [± 0.04] | −8.78 | $1.17 \times 10^{-16}$ | Very high certainty | |
| | | 50th | After November 2017 | | 0.62 [± 0.44] | 2.79 | $2.62 \times 10^{-3}$ | High certainty | 6.85 |
| | | 95th | Before November 2017 | | −0.26 [± 0.08] | −6.09 | $3.32 \times 10^{-9}$ | Very high certainty | |
| | | 95th | After November 2017 | | 1.6 [± 1.2] | 2.62 | $1.43 \times 10^{-3}$ | High certainty | 1.10x10-2 |
| São Paulo (21) | January 1998– December 2020 | 5th | Before March 2008 | 6.1 | −0.19 [± 0.21] | −1.8 | $7.30 \times 10^{-2}$ | Medium certainty | |
| | | 5th | After March 2008 | | 0.38 [± 0.27] | 2.81 | $9.18 \times 10^{-3}$ | High certainty | 5.54 |
| | | 50th | Before March 2008 | | −0.17 [± 0.23] | −1.51 | $1.33 \times 10^{-1}$ | Low certainty | |
| | | 50th | After March 2008 | | 0.31 [± 0.23] | 2.72 | $3.81 \times 10^{-3}$ | High certainty | 7.17 |
| | | 95th | Before March 2008 | | −0.05 [± 0.43] | −0.25 | $8.04 \times 10^{-1}$ | Very low certainty | |
| | | 95th | After March 2008 | | 0.43 [± 0.32] | 2.68 | $1.67 \times 10^{-2}$ | High certainty | 8.25x10-3 |
| Tololo (1) | December 1995– December 2021 | 5th | Before May 2006 | 3.6/4.2 | −0.04 [± 0.22] | −0.37 | $7.09 \times 10^{-1}$ | Very low certainty | |
| | | 5th | 2006–2014 | | 0.16 [± 0.16] | 1.98 | $2.64 \times 10^{-4}$ | Medium certainty | 4.94x10-2 |
| | | 5th | After April 2014 | | 0.33 [± 0.48] | 1.37 | $1.68 \times 10^{-1}$ | Low certainty | 1.72 |
| | | 50th | Before May 2006 | | 0.05 [± 0.12] | 0.88 | $3.82 \times 10^{-1}$ | Very low certainty | |
| | | 50th | 2006–2014 | | 0.29 [± 0.14] | 4.07 | $2.41 \times 10^{-2}$ | Very high certainty | 6.80x10-5 |
| | | 50th | After April 2014 | | 0.07 [± 0.31] | 0.44 | $7.39 \times 10^{-1}$ | Very low certainty | 6.63 |
| | | 95th | Before May 2006 | | −017 [± 0.28] | −1.17 | $2.41 \times 10^{-1}$ | Low certainty | |
| | | 95th | 2006–2014 | | 0.40 [± 0.19] | 4.16 | $4.55 \times 10^{-2}$ | Very high certainty | 4.85x10-5 |
| | | 95th | After April 2014 | | 0.04 [± 0.29] | 0.29 | $6.74 \times 10^{-1}$ | Very low certainty | 7.72 |

elevation sites, mainly in the Northern Hemisphere (Putero et al., 2023). In contrast, Ushuaia, located in a net ozone depletion zone in the southernmost region of South America (Adame et al., 2019), showed a decreasing ozone trend, with a low rate ($-0.07$ nmol mol$^{-1}$ yr$^{-1}$) and high certainty (SNR > 2.95). These trend estimations are consistent with previous work (Cooper et al., 2020).

## 3.3 Ozone change points

Several South American urban agglomerations exhibit similarities in terms of implementing measures to reduce air pollution by limiting the accelerated growth in motorized vehicles. In general terms, we note that these ozone precursor abatement measures have been implemented, ignoring the VOC-to-NO$_x$ ratio, suggesting that ozone increases once the VOC-limited regime is reached. Under this chemical regime (high VOC-to-NO$_x$ ratio), less NO is available to titrate O$_3$ due to fewer vehicle emissions, and VOC oxidations initiated by hydroxyl radicals (OH) efficiently convert NO to NO$_2$, the photolysis of which produces O$_3$ (Monks et al., 2015).

Moreover, biogenic VOCs are known as effective ozone precursors due to their high reactivity toward OH (Atkinson and Arey, 2003). Unfortunately, in South America, biogenic VOC measurements are derived from field campaigns conducted for short time periods; therefore, the direct influence of biogenic VOCs on ozone levels is beyond the scope of this study. However, since higher vegetation releases are expected to occur during months impacted by extreme temperatures and heat waves, their influence on the summer months is likely to be reflected in ozone trends. The latter, together with the extensive wildfires around the cities studied, could explain the occurrence of trend change points at some sites.

The ozone mixing ratios in Bogotá showed no evidence of reduction or increase during the last decade despite efforts to reduce primary pollutant emissions (Fig. 5a). Figure 5c and d also show the trends in CO and NO$_x$ and their change point with 95 % confidence interval to illustrate the low ozone sensitivity under this chemical regime, i.e., before and after the precursors' change point.

CO showed a decreasing trend between 2008 and 2014 ($-35$ nmol mol$^{-1}$ yr$^{-1}$ in the 50th percentile), attributable

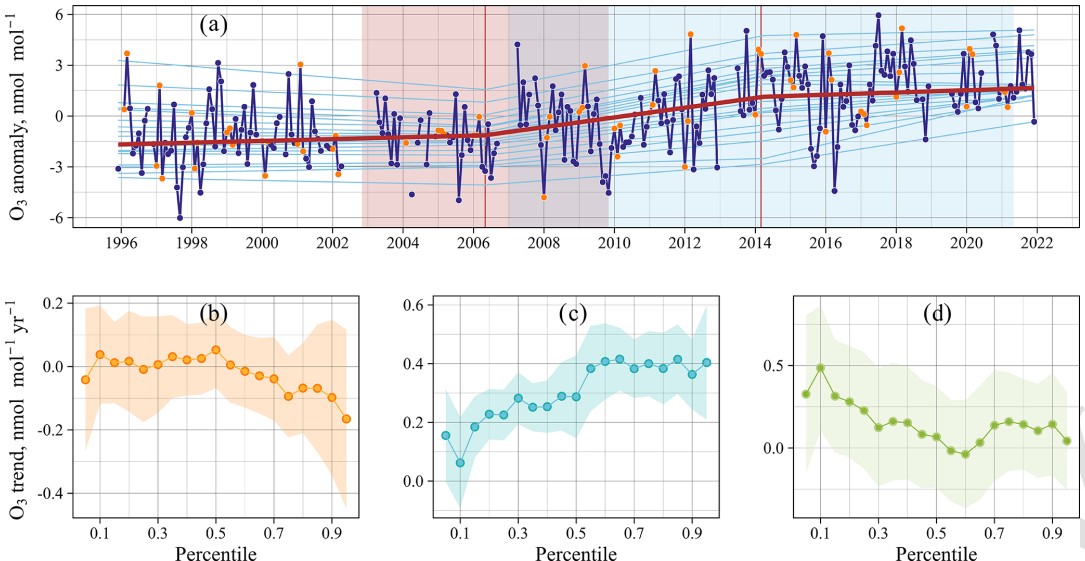

**Figure 4.** Percentile trends derived by quantile regression based on the monthly surface ozone **(a)** in Tololo. The orange dots in panel **(a)** indicate the first 3 months of every year for reference purposes. In panel **(a)**, the red line corresponds to the 50th percentile, and the light blue lines correspond to the remaining percentiles. Change points with 95 % confidence intervals are represented by a vertical red line and shaded red (first change point) and light blue (second change point). Panels **(b)**, **(c)** and **(d)** show the percentile trends of quantile regression from the 5th to 95th percentiles at 5 percentile intervals in Tololo before the first change point, between the change points and after the last change point.

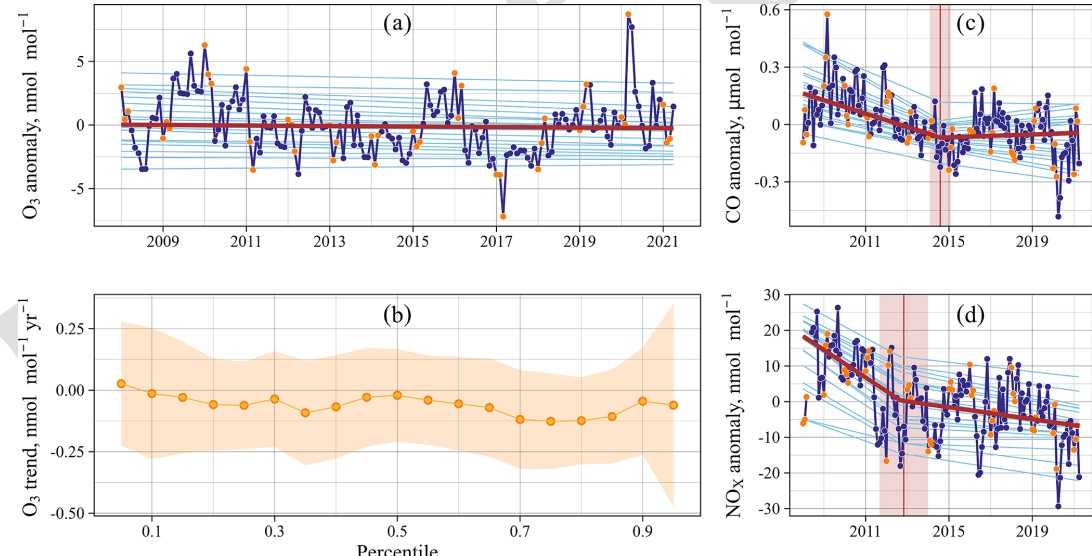

**Figure 5.** Percentile trends derived by quantile regression based on the monthly surface ozone **(a)**, carbon monoxide **(c)** and nitrogen oxide anomalies **(d)** in Bogotá. The orange dots in panels **(a)**, **(c)** and **(d)** indicate the first 3 months of every year for reference purposes. In panels **(a)**, **(c)** and **(d)**, the red line corresponds to the 50th percentile, and the light blue lines correspond to the remaining percentiles. Change points (when any) with 95 % confidence intervals are represented by a vertical red line and shaded red, respectively. Panel **(b)** shows the percentile trends of quantile regression from the 5th to 95th percentiles at 5 percentile intervals in Bogotá.

to improvements in the quality of fuels, renewal of vehicle fleets with better emission standards and substitution of coal with natural gas at industrial facilities. From August 2014 onwards, the trend changed to an increase in CO (3.6 nmol mol$^{-1}$ yr$^{-1}$ in the 50th percentile), particularly in

the high percentiles (> 45th). This could be explained by the increase in the motorcycle fleet of the city, whose emission standard is lower than that of passenger cars (Rojas et al., 2023). In contrast, NO$_x$ decreased rapidly between 2008 and 2012 (−3.9 nmol mol$^{-1}$ yr$^{-1}$ in the 50th percentile)

and more slowly after the change point in November 2012 ($-0.84$ nmol mol$^{-1}$ yr$^{-1}$ in the 50th percentile).

In Quito, the median ozone trends slowly decreased until April 2011, at a rate of $-0.22$ nmol mol$^{-1}$ yr$^{-1}$ (high certainty). After the change point, the trend reliability was very low given the change point confidence interval ($\pm$ [August 2008, December 2013]. Therefore, there was no evidence of a change in the ozone trend (Fig. 6a). In turn, the relationship between NO$_2$ and CO with ozone was more evident before the precursors' change point when the increase in the NO$_2$ anticorrelated with ozone. This behavior allows us to infer a higher ozone destruction due to NO titration (note that NO is not measured in Quito).

The change points of CO and NO$_2$ detected in April 2013 $\pm$ [August 2012, December 2013] and July 2013 $\pm$ [May 2012, September 2014] overlapped with the ozone change point, indicating a shift from higher ozone inhibition observed around the precursors' change point to a more uncertain regime (Fig. 6). Also, the reliability of the CO and NO$_2$ trends shows low or very low certainty after 2013 (Fig. 6a).

Ozone in Santiago decreased for nearly 2 decades due to public policies mainly focusing on curbing particulate matter in cold seasons. Thus, particles have decreased over time (Jorquera, 2020), while the benefits of those policies turned ineffective in curbing ozone during warm seasons, particularly in the last decade (Seguel et al., 2020). Figure 7a shows the percentile trend based on monthly ozone anomalies for the entire city of Santiago. The median trend showed that between 1997 and 2017, ozone decreased at a rate of $-0.19$ nmol mol$^{-1}$ yr$^{-1}$ (very high certainty). After November 2017 $\pm$ [March 2017, July 2018], an increase of 0.62 nmol mol$^{-1}$ yr$^{-1}$ has been observed for the last 5 years (very high certainty). Figure 7b and c show the percentile trends before and after the change points. Up to 2017, the highest percentiles ($\geq$ 80th) exhibited the most significant decreasing trends ($> -0.23$ nmol mol$^{-1}$ yr$^{-1}$). In contrast, after the 2017 change point, the highest percentiles ($\geq$ 80th) showed greater increasing trends ($> 0.91$ nmol mol$^{-1}$ yr$^{-1}$). In other words, until 2017, the policies lowered the highest ozone percentiles.

Ozone sensitivity to NO$_x$ and CO is observed in Fig. 7. The carbon monoxide anomalies clearly improved after 2010, resulting from the massive introduction of natural gas in the Santiago Metropolitan Region in 2009 (Mena-Carrasco et al., 2012). Notably, the cold-month anomalies (April–September) decreased drastically. However, the warm-month anomalies (October–March) started to increase again after 2017. In turn, the median trend remained flat after 2006. In contrast, the nitrogen oxide mixing ratio decreased after 2017 ($-1.94$ nmol mol$^{-1}$ yr$^{-1}$). Thus, the CO-to-NO$_x$ ratio increase after the NO$_x$ change point provides a favorable scenario for efficient ozone formation. The latter observation is also evident in the NO$_2$-to-NO$_x$ ratio, which typically decreases in winter due to higher NO primary emissions but increases in summer due to VOC oxidations. These oxidations involve reactions of hydroperoxyl (HO$_2$) and alkyl peroxy radicals (RO$_2$) with nitric oxide (NO), leading to the production of NO$_2$. Figure 8 illustrates the intense photochemical activity in Santiago, with a notably higher NO$_2$-to-NO$_x$ ratio ($> 0.6$), particularly during the warm season (October to March). In contrast, Bogotá exhibits a lower NO$_2$-to-NO$_x$ ratio, seldom exceeding 0.5, except during periods such as the COVID-19 lockdowns. This pattern suggests that NO$_2$ formation is typically suppressed under Bogotá's current chemical conditions.

Over the last 2 decades, NO$_x$ and CO mixing ratios have successfully decreased in the São Paulo Metropolitan Area, despite the growth in the vehicular fleet, due to regulations aimed at abating motorized vehicle emissions (Fig. 9). The Brazilian Program for the Control of Air Pollution Emissions of Motor Vehicles has contributed to decreasing vehicular emissions in Brazil. In this regard, reductions in CO and NO$_x$ emissions from light- and heavy-duty vehicles varying between $-2.9$ % yr$^{-1}$ and $-5.1$ % yr$^{-1}$ between 2001 and 2018 have been accomplished (Nogueira et al., 2021).

Figure 9 depicts a consistent decrease in CO over the period, marked by a change point in September 2008 $\pm$ [September 2006, September 2010], coinciding with the ozone change point detected in March 2008 $\pm$ [September 2006, September 2009]. From 2008 to 2020, ozone increased by approximately 4 nmol mol$^{-1}$ in the 50th percentile (Table 3). Notably, many higher ozone anomalies occurred in the warmer months (January–February) and were more frequent after the last NO$_x$ change point detected in June 2013 $\pm$ [April 2011, August 2015]. Accordingly, the ozone trends in the 90th and 95th percentiles increased at rates of 0.44 and 0.43 nmol mol$^{-1}$ yr$^{-1}$, respectively, compared with those in the lower 80th–15th percentiles (Fig. 9c). The intensification of photochemical activity is evident in Fig. 8, where the NO$_2$-to-NO$_x$ ratio exceeded 0.6 after the 2013 NO$_x$ change point.

## 3.4 Extreme ozone anomalies

As has been widely characterized in the literature, ozone anomalies and their precursors clearly show the effect of pandemic confinement in 2020 in many South American cities. In Santiago, positive ozone anomalies occurred in April (4.1 nmol mol$^{-1}$) and October (4.9 nmol mol$^{-1}$) 2020; in Quito in March (3.5 nmol mol$^{-1}$), April (4.3 nmol mol$^{-1}$) and October (3.2 nmol mol$^{-1}$); and in São Paulo in April (7.3 nmol mol$^{-1}$) and September (8.1 nmol mol$^{-1}$).

In turn, in Bogotá, the emission reductions during the COVID-19 lockdowns, combined with the transport of precursors emitted during biomass fires in the Colombian–Venezuelan plains (Ballesteros-González et al., 2020), contributed to positive extreme ozone anomalies in March (8.7 nmol mol$^{-1}$) and April (7.7 nmol mol$^{-1}$) (Sokhi et al., 2021; Mendez-Espinosa et al., 2019). During this event, the

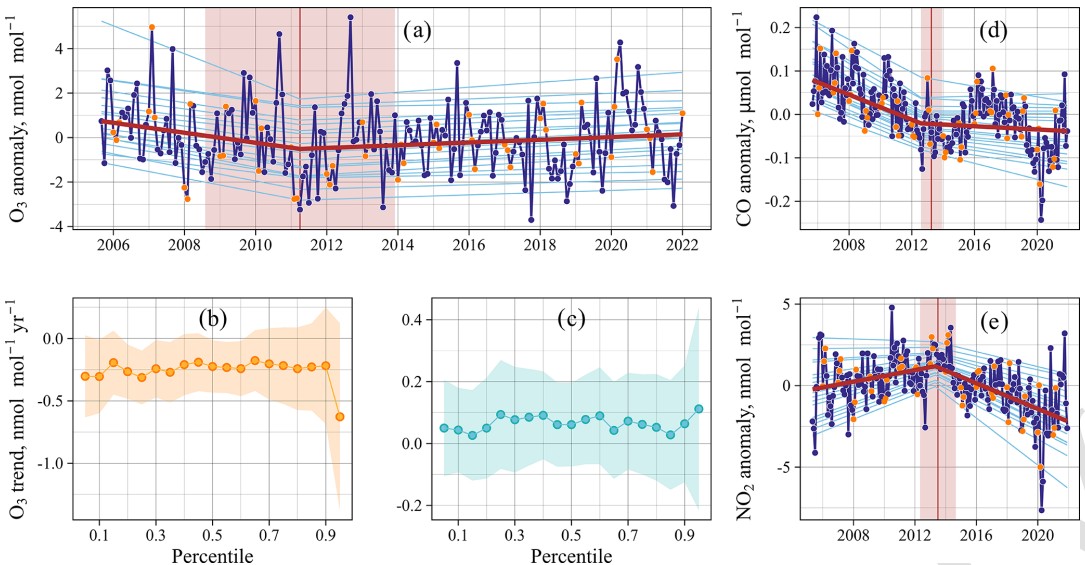

**Figure 6.** Percentile trends derived by quantile regression based on the monthly surface ozone **(a)**, carbon monoxide **(d)** and nitrogen dioxide anomalies **(e)** in Quito. The orange dots in panels **(a)**, **(d)** and **(e)** indicate the first 3 months of every year for reference purposes. In panels **(a)**, **(d)** and **(e)**, the red line corresponds to the 50th percentile, and the light blue lines correspond to the remaining percentiles. Change points (when any) with 95 % confidence intervals are represented by a vertical red line and shaded red, respectively. Panels **(b)** and **(c)** show the percentile trends of quantile regression from the 5th to 95th percentiles at 5 percentile intervals before and after the change point in Quito.

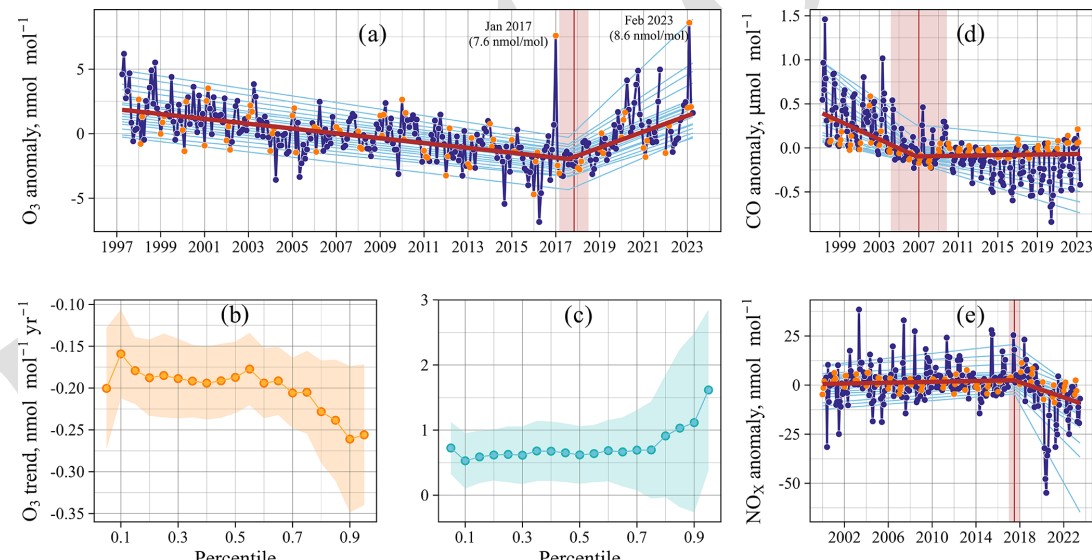

**Figure 7.** Percentile trends derived by quantile regression based on monthly surface ozone **(a)**, carbon monoxide **(d)** and nitrogen oxide anomalies **(e)** in Santiago. The orange dots in panels **(a)**, **(d)** and **(e)** indicate the first 3 months of every year for reference purposes. In panels **(a)**, **(d)** and **(e)**, the red line corresponds to the 50th percentile, and the light blue lines correspond to the remaining percentiles. Change points (when any) with 95 % confidence intervals are represented by a vertical red line and shaded red, respectively. Panels **(b)** and **(c)** show the percentile trends of quantile regression from the 5th to 95th percentiles at 5 percentile intervals in Santiago before and after the change point.

highest ozone mixing ratios were associated with clear skies, notable thermal inversion, stagnant conditions and westerly winds in the afternoon, which trapped the air against the eastern mountains that border the city.

Also, the long-lasting drought that has affected the west coast of subtropical South America, coupled with extreme weather configurations, has resulted in conditions capable of modifying the fire regime in southern-central Chile (mid-latitudes) (González et al., 2018). The meteorological pattern

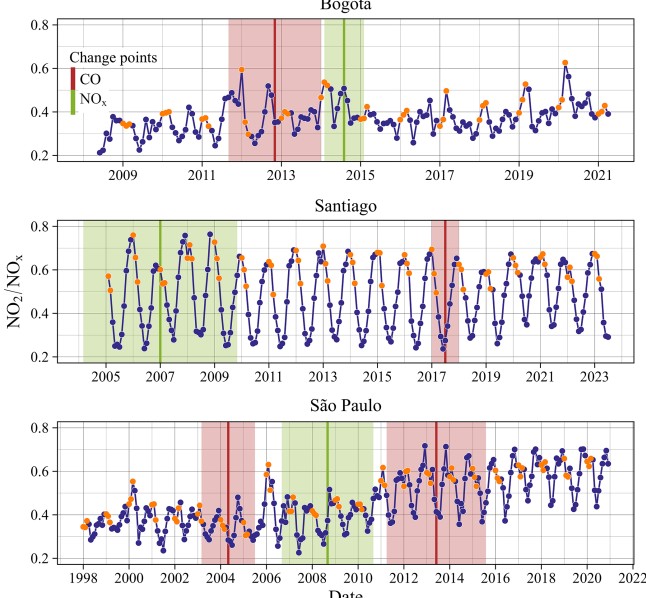

**Figure 8.** $NO_2$-to-$NO_x$ ratio based on surface monthly means. The orange dots indicate the first 3 months of every year for reference purposes. Change points with 95 % confidence intervals are represented by a vertical red line (green line) for nitrogen oxides (carbon monoxide) and shaded red (green).

includes extreme temperatures (not necessarily heat waves) and low relative humidity levels with a high-pressure system that provides atmospheric stability, i.e., descending air masses and warm and dry wind, which cause fire propagation. In this regard, extreme positive ozone anomalies were observed in January 2017 ($7.6\,nmol\,mol^{-1}$) and February 2023 ($8.6\,nmol\,mol^{-1}$), caused by ozone and precursors transported from areas affected by intense wildfires (Fig. 7a). Figure B1 (Appendix B) shows events in early February 2023 characterized by many active fires whose impact was quantifiable by satellite observations. During this period, high CO ($1.23\,\mu mol\,mol^{-1}$) and MDA8 ozone levels ($117\,nmol\,mol^{-1}$) were observed in Santiago and locations to the south in early February. In contrast, the increase in nitrogen oxides was less than that observed for CO and $O_3$, which may be due to the rapid conversion of nitrogen oxides into other more oxidized forms, such as particulate nitrate, during plume transport (Juncosa Calahorrano et al., 2021).

In Quito, warmer summers, whose maximum values are typically reached between August and September, together with the occurrence of wildfires in the surrounding woods of the city, have also produced high ozone mixing ratios. Although not all these events were sufficiently extended in time to produce significantly high monthly anomalies, the September 2015 anomaly coinciding with a fire is notable, as shown in Fig. 6a.

In the case of São Paulo, biomass burning is typically greater in September. This period marks the end of the dry season, when most air masses move from the northern and central-western regions of Brazil toward the south (São Paulo). However, with few exceptions, September is not the month with the highest anomalies. Indeed, many of the extreme ozone anomalies observed after the trend change point occurred in months in which long heat waves have been reported (Valverde and Rosa, 2023): February 2012 (11 d), October 2014 (10 d), January 2015 (13 d) and January 2019 (Fig. 9a).

## 4 Conclusions

Short-term (MDA8) and long-term (peak-season) exposure metrics calculated for the present day (2017–2021) revealed differences between the tropics and extratropics in South America. The tropical cities of Bogotá and Quito attained lower ozone exposure levels than the large extratropical cities of Santiago in Chile and São Paulo in Brazil. Factors such as convection and vertical mixing within a convective tropical setting could explain, in part, the lower levels of ozone observed in Quito and Bogotá. Medium-sized cities located downwind of Santiago and São Paulo also showed high ozone exposure levels.

Santiago and São Paulo contained receptor sites and urban subdivisions with positive trends and high certainty after the detected change points (mainly after 2010). We attributed these observed ozone trends to a greater decrease in nitrogen oxides than in carbon monoxide, which resulted in chemical regimes that efficiently convert nitric oxide into nitrogen dioxide. Quito (after a change point) and Bogotá showed no evidence of variations in ozone trends.

The relatively greater reduction in nitrogen oxides during the COVID-19 pandemic (mostly limited to 2020) combined with warmer summers and intense wildfires in the region produced extreme positive ozone anomalies capable of increasing the highest percentiles. The 95th (50th) percentile trends in Santiago and São Paulo were $1.6\,nmol\,mol^{-1}\,yr^{-1}$ ($0.62\,nmol\,mol^{-1}\,yr^{-1}$) and $0.43\,nmol\,mol^{-1}\,yr^{-1}$ ($0.31\,nmol\,mol^{-1}\,yr^{-1}$) over 5 and 12 years, representing an accumulated CE3 of $9\,nmol\,mol^{-1}$ ($3\,nmol\,mol^{-1}$) and $6\,nmol\,mol^{-1}$ ($4\,nmol\,mol^{-1}$), respectively. Therefore, extreme positive ozone anomalies in large urban agglomerations, whose ozone production regime is mostly VOC limited, constitute a regional challenge regarding ozone precursor mitigation and adaptation to warmer temperatures and new fire regimes.

Tololo is a valuable background site on the west coast of subtropical South America that showed signs of increased ozone. The ozone increase observed between 2006 and 2014 is notable, which, from a broader perspective, warns regarding changes in the ozone baseline.

These results advance the current knowledge described in the literature that addresses ozone trends in South America, which roughly provides surface ozone trends but over-

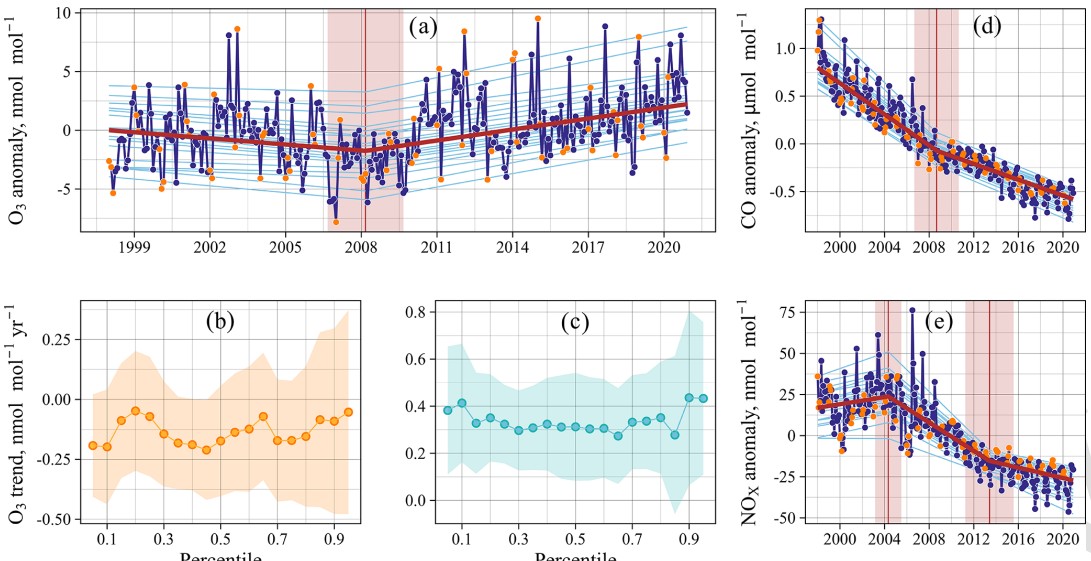

**Figure 9.** Percentile trends derived by quantile regression based on the monthly surface ozone **(a)**, carbon monoxide **(d)** and nitrogen oxide anomalies **(e)** in São Paulo. The orange dots in panels **(a)**, **(d)** and **(e)** indicate the first 3 months of every year for reference purposes. In panels **(a)**, **(d)** and **(e)**, the red line corresponds to the 50th percentile, and the light blue lines correspond to the remaining percentiles. Change points (when any) with 95 % confidence intervals are represented by a vertical red line and shaded red, respectively. Panels **(b)** and **(c)** show the percentile trends of quantile regression from the 5th to 95th percentiles at 5 percentile intervals in São Paulo before and after the change point.

looks change point attribution. However, the drawbacks of this study include the large unmonitored areas in the region, while an improvement for future research requires elevating the quality assurance–quality control procedures to the same level, in each nation first, and then for South America. This action could mean a potential addition of up to 70 ozone time series (49 %) rejected due to the controls we performed. Additionally, the absence of systematic monitoring of volatile organic compounds makes it difficult to determine the contributions of anthropogenic and biogenic reactive species to ozone trends.

Finally, our results revealed signs of a climate penalty for ozone in South America, derived from more favorable meteorological conditions for wildfire propagation in Chile and extensive heat waves in southern Brazil. In these regions' urban environments, the increase in ozone poses the highest risk. These observation-based results provide new evidence for comparison with trends in the free troposphere based on reanalysis, satellite observation and numerical simulation data.

## Appendix A: Year of the last detected ozone trend change point

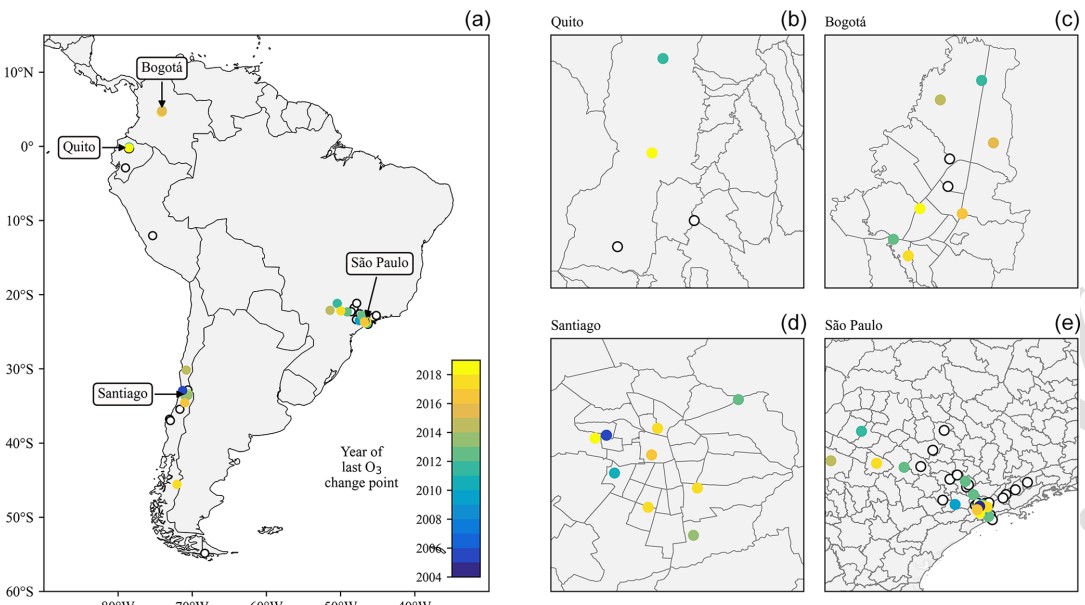

**Figure A1.** Year of the last detected ozone trend change point in South American monitoring stations **(a)**. Panels **(b)**–**(e)** focus on Bogotá, Quito, Santiago and São Paulo. The white dots denote stations with no change point detected.

## Appendix B: Ozone and ozone precursors' time series during 2023 wildfires in central Chile

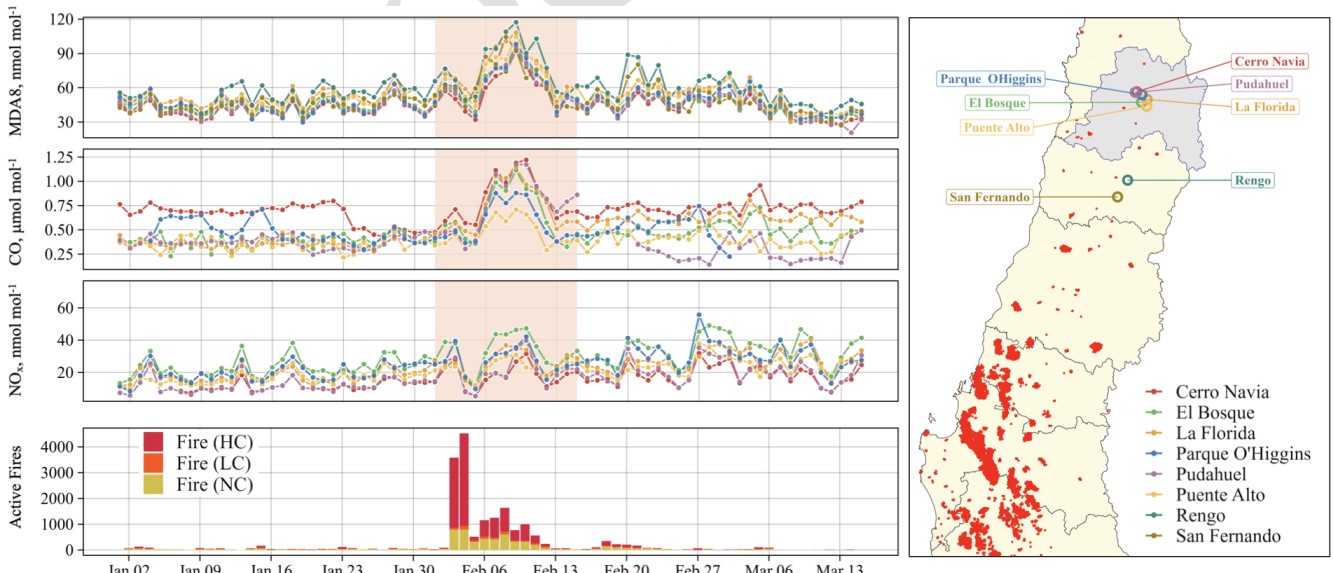

**Figure B1.** Time series of the maximum daily 8 h average (MDA8) ozone and its precursors carbon monoxide and nitrogen oxides for the summer of 2023. The lower panel shows the daily (24 h) active fires obtained from MODIS satellite products at 1 km resolution (MOD13A1). Colored bars indicate confidence (high, nominal and low) based on infrared radiation emission. The map (left panel) shows the active fires during February in central Chile. Santiago Metropolitan Region is indicated in grey on the map.

**Data availability.** Datasets used in this research are available in the TOAR-II database at https://toar-data.org [TS7]. Active fires can be downloaded at https://modis-fire.umd.edu/ [TS8]. Carbon monoxide and nitrogen oxide surface data (Fig. B1) can be downloaded at https://sinca.mma.gob.cl [TS9]. GAW data can be downloaded at https://ebas.nilu.no/ [TS10].

[TS11]We obtained information about active fires from the Terra Moderate Resolution Imaging Spectroradiometer (MODIS) Thermal Anomalies and Fire Daily (MOD14A1) satellite product (https://doi.org/10.5067/MODIS/MOD14A1.061, Giglio and Justice, 2021). The product includes a 1 km gridded fire mask providing information about the confidence level (low, nominal or high) of fire activity possibly detected in each grid cell over the 24 h compositing period. Using MOD14A1 grid cells within Chilean continental territory from 33 to 39° S, we derived daily time series of the total number of pixels with active fires for each confidence level.

**Supplement.** The supplement related to this article is available online at: https://doi.org/10.5194/acp-24-1-2024-supplement. [TS12]

**Author contributions.** RJS, NYR, TN, MC and YE: conceptualization. RJS, LC and CO: methodology. LC, TN, MC and MGC: data curation. RJS, LC, CO and TCE: formal analysis. RJS: writing (original draft preparation). All authors: writing (review and editing).

**Competing interests.** The contact author has declared that none of the authors has any competing interests.

**Special issue statement.** This article is part of the special issue "Tropospheric Ozone Assessment Report Phase II (TOAR-II) Community Special Issue (ACP/AMT/BG/GMD inter-journal SI)". It is a result of the Tropospheric Ozone Assessment Report, Phase II (TOAR-II, 2020–2024). [TS13]

**Acknowledgements.** This research was partially supported by ANID/FONDAP/1522A000 and the supercomputing infrastructure of the NLHPC (ECM-02). We acknowledge the use of data from the following public networks: Quito Air Quality Network and [CE4]Secretariat of the Environment.

**Financial support.** This research has been supported by the NAME OF FUNDER (grant no. GRANT AGREEMENT NO). [TS14]

**Review statement.** This paper was edited by Stefano Galmarini and reviewed by two anonymous referees.

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

## Remarks from the language copy-editor

CE1    Please note slight edit to this affiliation.

CE2    Please check this table carefully as any changes will not show due to conversion issues.

CE3    Please check; there seems to be a word missing here.

CE4    Please confirm whether this is the correct term. Should it be "Ministry"? Which country is it affiliated to?

## Remarks from the typesetter

TS1    Please provide department if possible.

TS2    Please provide department.

TS3    The composition of Figs. 1–3, 4–7, 9, and A1 has been adjusted to our standards.

TS4    This reference is not in the reference list. Please add it.

TS5    This reference is not in the reference list. Please add it.

TS6    Please confirm deletion of "et al.".

TS7    Please provide a direct link to the data set and, if possible, a DOI instead of a URL. In any case, please provide a reference list entry including creators, title, repository/publisher, and date of last access.

TS8    Please provide a direct link to the data set and, if possible, a DOI instead of a URL. In any case, please provide a reference list entry including creators, title, repository/publisher, and date of last access.

TS9    Please provide a direct link to the data set and, if possible, a DOI instead of a URL. In any case, please provide a reference list entry including creators, title, repository/publisher, and date of last access.

TS10    Please provide a direct link to the data set and, if possible, a DOI instead of a URL. In any case, please provide a reference list entry including creators, title, repository/publisher, and date of last access.

TS11    Please note that this information has been moved to the data availability section.

TS12    Please send a new supplement as a *.pdf without the title, authors, correspondence author, etc. as we will generate a supplement title page during publication (with a citation including the DOI), which will contain this information.

TS13    Please confirm.

TS14    Please note that there is funding information given in the acknowledgements but you have not indicated any funding upon manuscript registration. Therefore, we were not able to complete the financial support statement. Please fill the missing information and double-check your acknowledgements to see whether repeated information can be removed from the acknowledgement. Thanks.

TS15    Please ensure that any data sets and software codes used in this work are properly cited in the text and included in this reference list. Thereby, please keep our reference style in mind, including creators, titles, publisher/repository, persistent identifier, and publication year. Regarding the publisher/repository, please add "[data set]" or "[code]" to the entry (e.g. Zenodo [code]).

TS16    Please provide page range or article number.

TS17    Please provide page range or article number.

TS18    Please provide page range or article number.

TS19    Please provide page range or article number.

TS20    Please provide page range or article number.

TS21    Please provide page range or article number.

TS22    Please provide date of last access.

TS23    Please provide publisher.

TS24    Please provide page range or article number.

TS25    Please provide page range or article number.

TS26    Please check if "Sci. Rep." or "Sci. Rep.-UK" is meant here.

TS27    Please provide page range or article number.

TS28    This reference is not cited in the text. Please check.

TS29    Please provide page range or article number.

TS30    Please provide page range or article number.

TS31    Please provide page range or article number.

TS32    Please provide page range or article number.

TS33    Please provide page range or article number.

TS34    Please provide page range or article number.

TS35    Please provide page range or article number.

TS36 Please provide journal, volume number and page range or article number.

TS37 Please provide volume number and page range or article number.

TS38 Please provide page range or article number.

TS39 Please provide page range or article number.

TS40 Please provide page range or article number.

TS41 Please provide page range or article number.

TS42 Please provide publisher.

TS43 Please provide page range or article number.

TS44 Please provide journal, volume number and page range or article number.

TS45 Please provide page range or article number.

TS46 Please provide page range or article number.

TS47 Please provide page range or article number.

TS48 Please provide journal, volume number and page range or article number.

TS49 Please provide page range or article number.

TS50 Please provide page range or article number.

TS51 Please provide page range or article number.

TS52 Please provide volume number and page range or article number.

TS53 Please provide page range or article number.

TS54 Please provide page range or article number.

TS55 Please provide page range or article number.

TS56 Please provide page range or article number.

TS57 Please provide volume number and page range or article number.

TS58 Please provide page range or article number.

TS59 Please provide name.

TS60 Please provide page range or article number.

TS61 Please provide page range or article number.

TS62 Please provide journal, volume number and page range or article number.

TS63 Please provide page range or article number.

TS64 Please provide a persistent identifier (ISBN or DOI preferred).

https://doi.org/10.5194/acp-24-1-2024