# Peer review of "Changes in South American Surface Ozone Trends: Exploring the Influences of Precursors and Extreme Events"

_EGUsphere, 2024_

## Community Comment (CC1)

Comments by Owen R. Cooper (TOAR Scientific Coordinator of the Community Special Issue) on:

**Changes in South American Surface Ozone Trends: Exploring the Influences of Precursors and Extreme Events**

Seguel, R. J., Castillo, L., Opazo, C., Rojas, N. Y., Nogueira, T., Cazorla, M., Gavidia-Calderón, M., Gallardo, L., Garreaud, R., Carrasco-Escaff, T., and Elshorbany, Y.

EGUsphere [preprint], https://doi.org/10.5194/egusphere-2024-328, 2024.
Discussion started: 12 Feb 2024;  Discussion closes March 25, 2024

This review is by Owen Cooper, TOAR Scientific Coordinator of the TOAR-II Community Special Issue. I, or a member of the TOAR-II Steering Committee, will post comments on all papers submitted to the TOAR-II Community Special Issue, which is an inter-journal special issue accommodating submissions to six Copernicus journals:  ACP (lead journal), AMT, GMD, ESSD, ASCMO and BG. The primary purpose of these reviews is to identify any discrepancies across the TOAR-II submissions, and to allow the author teams time to address the discrepancies.  Additional comments may be included with the reviews. While O. Cooper and members of the TOAR Steering Committee may post open comments on papers submitted to the TOAR-II Community Special Issue, they are not involved with the decision to accept or reject a paper for publication, which is entirely handled by the journal's editorial team.

**General Comments:**

TOAR-II has produced two guidance documents to help authors develop their manuscripts so that results can be consistently compared across the wide range of studies that will be written for the TOAR-II Community Special Issue.  Both guidance documents can be found on the TOAR-II webpage: https://igacproject.org/activities/TOAR/TOAR-II

*The TOAR-II Community Special Issue Guidelines*:   In the spirit of collaboration and to allow TOAR-II findings to be directly comparable across publications, the TOAR-II Steering Committee has issued this set of guidelines regarding style, units, plotting scales, regional and tropospheric column comparisons, tropopause definitions and best statistical practices.

*Guidance note on best statistical for TOAR analyses*:  The aim of this guidance note is to provide recommendations on best statistical practices and to ensure consistent communication of statistical analysis and associated uncertainty across TOAR publications. The scope includes approaches for reporting trends, a discussion of strengths and weaknesses of commonly used techniques, and calibrated language for the communication of uncertainty. Table 3 of the TOAR-II statistical guidelines provides calibrated language for describing trends and uncertainty, similar to the approach of IPCC, which allows trends to be discussed without having to use the problematic expression, "statistically significant".

**Major Comments:**

The authors have provided the first continent-wide overview of surface ozone across South America, based on all available observations. This is an important topic and a welcome addition to the TOAR-II Community Special Issue.  The trend analysis is extremely well done, and the authors have done a very good job of following the recommendations from the "Guidance note on best statistical for TOAR analyses". In particular the use of the TOAR vector approach for visualizing trends and the use of the TOAR color table makes it very easy to compare these new TOAR results to previous TOAR studies.

Overall the findings are consistent with the papers from TOAR-I and with the papers submitted so far to the TOAR-II Community Special Issue. A paper that is currently under review with the TOAR-II Community Special Issue reports boundary layer and free tropospheric ozone observations across the tropics based on IAGOS commercial aircraft and ozonesondes. It would be helpful if the authors could briefly discuss how their findings are relevant to these other new results, or how the ozone values in the free troposphere (reported by Gaudel et al., 2024) might affect the surface sites:

Gaudel, A., Bourgeois, I., Li, M., Chang, K.-L., Ziemke, J., Sauvage, B., Stauffer, R. M., Thompson, A. M., Kollonige, D. E., Smith, N., Hubert, D., Keppens, A., Cuesta, J., Heue, K.-P., Veefkind, P., Aikin, K., Peischl, J., Thompson, C. R., Ryerson, T. B., Frost, G. J., McDonald, B. C., and Cooper, O. R.: Tropical tropospheric ozone distribution and trends from in situ and satellite data, EGUsphere [preprint], https://doi.org/10.5194/egusphere-2023-3095, 2024.

Mixing ratios are reported in units of ppbv, however, Copernicus journals require units of nmol mol$^{-1}$.

Line 358
Tololo is mentioned in the Conclusions and acknowledged as being a valuable monitoring station, but it is not mentioned much in the main text. A figure showing the full Tololo time series, along with its change points (e.g. Figure 7a), would be very helpful and would clearly illustrate the shifts in background ozone.

Data Availability statement:
Please provide additional details that will allow the reader to find the data.
1) provide a link to the TOAR-II surface ozone database
2) provide a link to the MODIS data
3) Figure B1 shows observations of CO and NOx. Are these data available from the TOAR database?
4) to acknowledge the WMO GAW program, please also provide a link to the location where GAW data can be downloaded: https://ebas.nilu.no/

**Minor Comments:**

Abstract: The way the first sentence is written, the subject of the sentence is "trends" and not "ozone". Therefore, the word "precursors" refers to trends, and not ozone. A better way to phrase the sentence is:
"In this study, trends of 21st-century ground-level ozone and ozone precursors were examined across South America, an understudied region where trend estimates have rarely been comprehensively addressed."

Introduction, first paragraph:
Section 2.2.5.3 in Chapter 2 of IPCC AR6 (Gulev et al., 2021) provides a concise summary of global tropospheric ozone trends, based on the TOAR findings. I recommend that the IPCC findings be used as the starting point for the trend discussion in the submitted manuscript:
"Since the mid-1990s, free tropospheric ozone has increased by 2–7% per decade in most regions of the northern mid-latitudes, and 2–12% per decade in the sampled regions of the northern and southern tropics (high confidence). Limited coverage by surface observations precludes identification of zonal trends in the SH, while observations of tropospheric column ozone indicate increases of less than 5% per decade at southern mid-latitudes (medium confidence)."

Gulev, S.K., P.W. Thorne, J. Ahn, F.J. Dentener, C.M. Domingues, S. Gerland, D. Gong, D.S. Kaufman, H.C. Nnamchi, J. Quaas, J.A. Rivera, S. Sathyendranath, S.L. Smith, B. Trewin, K. von Schuckmann, and R.S. Vose, 2021: Changing State of the Climate System. In Climate Change 2021: The Physical Science Basis.

Contribution of Working Group I to the Sixth Assessment Report of the Intergovernmental Panel on Climate Change [Masson-Delmotte, V., P. Zhai, A. Pirani, S.L. Connors, C. Péan, S. Berger, N. Caud, Y. Chen, L. Goldfarb, M.I. Gomis, M. Huang, K. Leitzell, E. Lonnoy, J.B.R. Matthews, T.K. Maycock, T. Waterfield, O. Yelekçi, R. Yu, and B. Zhou (eds.)]. Cambridge University Press, Cambridge, United Kingdom and New York, NY, USA, pp. 287–422, doi:10.1017/9781009157896.004

Line 48
When summarizing ozone standards, the averaging time should also be given. For example, do the values 51-71 ppbv refer to the maximum daily 8-hour average?

Line 59
Please specify that GAW is a WMO program.

Line 104
The WHO AQ guidelines report is missing from the list of references:
WHO global air quality guidelines. Particulate matter (PM2.5 and PM10), ozone, nitrogen dioxide, sulfur dioxide and carbon monoxide. Geneva: World Health Organization; 2021. Licence:
CC BY-NC-SA 3.0 IGO.

Line 220
Ozone increased at Tololo from 2006 to 2014 by about 2.3 ppbv.  The discussion seems to imply that this increase of ozone is due to the increase of methane over the period 2006-2014.  The observed methane increase from 2006 to 2014 (according to NOAA GML: https://gml.noaa.gov/ccgg/trends_ch4/) was about 50 ppbv, or about 3%. While methane drives the background increase in ozone, are there any modelling results that can support this suggestion? A recent submission to the TOAR-II Community Special Issue (Nalam et al., 2024) calculates the change in global surface ozone due to methane increases over the period 2000-2018.  For the period 2006-2014, Figure 7b, indicates an ozone increase of no more than 1 ppbv.

Nalam, A., Lupascu, A., Ansari, T., and Butler, T.: Regional and sectoral contributions of NOx and reactive carbon emission sources to global trends in tropospheric ozone during the 2000–2018 period, EGUsphere [preprint], https://doi.org/10.5194/egusphere-2024-432, 2024.

Table 1
Please check the greater-than-or-equal-to symbols in this table against the original Table 3 in the "Guidance note on best statistical for TOAR analyses". Many of these symbols don't match the original table.

Figure B1
For those of us not highly familiar with the geography of Chile, it's not clear that the cluster of observing stations near the top of the map is located in the Santiago urban area. Can this be indicated on the map?

---

## Author Comment (AC1)

**Author Comments (egusphere-2024-328)**

Manuscript title: Changes in South American Surface Ozone Trends: Exploring the Influences of Precursors and Extreme Events

We have carefully read the referee and community comments. We greatly appreciate their quality and constructiveness. Accordingly, we have addressed each comment and incorporated the suggested changes in the new version of the manuscript. The referee and community comment revisions are addressed below.

**Referee Comment (RC1)**

**General comments:**

The manuscript presents a comprehensive analysis of the distribution and trends in long-term ozone and ozone precursor observations in cities and background locations in South America. While mainly European and North American ozone records are extensively studied and its interpretation can be found in the peer-reviewed literature, such studies are rather limited for South America. Therefore, the present manuscript provides a valuable contribution to the understanding of ozone trends in this less studied region.

The determination of the trends and the change points of the trends is sound. I would have just liked to learn more about the underlying data (analytical methods, quality control, screening, …) since the quality of the data is a crucial requirement for the analysis.

(**Answer**: Please note that this comment is addressed in the specific comments).

The paper will fit well into the TOAR-II Community Special Issue. See below a few specific comments that should be addressed prior to publication.

**Specific comments:**

Line 22-24: not clear which metric the numbers are referring to.

**Answer**: Thank you for noticing this. We reframe as:

"*Additionally, the maximum daily 8-hour average (MDA8) and peak-season metrics were used to assess short- and long-term exposure levels, respectively, for present-day (2017-2021).*"

Line 25: reader does not know yet how short-term and log-term exposure levels are defined. Add some information from lines 101 ff.

**Answer**: The above correction clarifies that the metrics used to evaluate short- and long-term exposure correspond to the MDA8 and Peak Season, respectively. In addition, we are aware that we are not yet providing a definition of each metric (i.e., how exactly are those calculated). However, we believe that the main point here, in the abstract, is to inform that the short- and long-term exposures were evaluated in the paper.

Line 26: trends refer to which metric?

**Answer**: To clarify this point, we added, "We applied the quantile regression method based on monthly anomalies to estimate trends..."

Line 54: replace "a chemical regime […] has been established …" by "a chemical regime […] has been found …"

**Answer**: We made the change suggested.

Lines 76-79: this sentence reads like a part of the conclusions. Move it below?

**Answer**: Thank you for noticing this. Although the intention of this sentence is to provide a hypothesis rather than a premature conclusion, we agreed that we need some reword to ensure the intended meaning. We eliminated the part that read as conclusion and reframed it as follows: "*We propose that the precursor ratio (nitrogen oxides to VOC) largely determines the observed ozone trends in South American urban environments, while short-lived but increasingly recurrent extreme weather events (high temperatures, low relative humidity levels and moderate to high winds) may also impact ozone trends*".

Lines 83 ff.: add some brief description of the measurement techniques. All UV absorption for $O_3$? chemiluminescence for NO and $NO_2$? If measurements are done by regulatory networks, I assume that $NO_2$ was converted to NO prior to detection with heated surface (molybdenum) converters. It is known that these converters overestimate the $NO_2$ mole fractions, especially in rural areas. This contribution may change over time when the amount of oxidized nitrogen species decreases. CO measurements with NDIR?

**Answer**: We added: "*The ozone measurement principle is based on a UV absorption technique, $NO_X$ on chemiluminescence, where $NO_2$ is converted to NO by a molybdenum converter heated before detection and finally, CO is measured using an infrared absorption technique*".

Please note that every time series (shared in the TOAR-II database) utilized in this study includes a header specifying the measurement techniques.

We agreed that $NO_x$ measurements can be tricky. Below (in the next answer), it is possible to see how many $NO_x$ measurements were rejected from the original dataset obtained from the environmental agencies and according to our own quality control.

Lines 86-87: please elaborate on the data screening performed by the authors. How was drift (trends in the instruments' response, I suppose) and representativeness assessed? A 75% data coverage criterion is mentioned below. This should be added here. Did you also exclude other data such as outliers, periods with very little or very large noise, … or was the quality of the received data just good. Please add how many data/datasets were rejected prior to your analysis. Lines 189-191 provide some of this information. Still the "quality control test established in the methodology" remains unclear.

**Answer**: The following text, added in the manuscript, addressed these questions:

"*Nonetheless, the authors assessed the datasets by applying further checks: (1) We evaluated the completeness by considering time series with at least 75% of daily, monthly and annual data and three years of consecutive measurements; (2) for each time series, we assessed (expert judgment) the remaining data gaps and the instrument response after those gaps; (3) we check the integrity of the time series now based on monthly anomalies; and finally (4) precursors time series that do not measure ozone at the monitoring site were omitted. As a result of this process, we produced a homogenized dataset, which was utilized in this research and submitted to the database of the Tropospheric Ozone Assessment Report, phase II (TOAR-II).*"

The Table below shows the number of stations that were rejected after each of the additional quality control steps.

| Pollutant | Data obtained from Env. Agencies | Step 1 | Step 2 | Step 3 | Step 4 (Final data) |
|---|---|---|---|---|---|
| $O_3$ | 144 | 106 | 81 | 74 | |
| CO | 108 | 67 | 52 | 43 | 33 |
| NO | 130 | 93 | 55 | 38 | 33 |
| $NO_2$ | 147 | 98 | 78 | 68 | 58 |
| $NO_x$ | 122 | 92 | 75 | 63 | 52 |

As can be seen in the above Table, we did not use the data as it came. Regarding the outlier question, normally, environmental agencies filter outliers. However, some outliers can be eliminated after the process described above.

Concerning the representativity question, most of the monitoring stations utilized aim to protect human health. However, a few monitoring stations included in this study are in industrial zones. Therefore, we aggregated these stations as Industrial (e.g., Industrial São Paulo). We added: "*Such aggregation and subdivision operations were performed according to local expert judgment, thereby accounting for representativity (human health, baseline, industrial influence), altitude, topography and precursor sources.*"

Figure 2, caption: "The black dots denote the monitoring stations that do not meet the data quality criteria." I do not see any back dots.

**Answer**: Indeed, the dots are gray and not black. We modified the text accordingly.

Lines 163 ff.: did I get it right? You attribute the lower $O_3$ levels in Quito to intense vertical mixing that mixes (less $O_3$-rich) air from the free troposphere to the site. Is there no signature from stratospheric intrusions seen at this elevation?

**Answer**: To the best of our knowledge, there is no signature of frequent ozone stratospheric intrusions in this region. Regarding the previous statement, in Gaudel (2024), vertical profiles (Figure 2) show no ozone structures over South American tropics. In any case, we do not rule out the occurrence of isolated events.

We added: "*This finding is consistent with processes that occurred in the tropics such as higher ozone photolysis ( $\lambda \leq 336$ nm) promoted by the intense solar ultraviolet radiation, high humidity favoring water reaction with atomic oxygen ($O(^1D)$) and the strong convection producing upward airflow, resulting in short $O_3$ lifetime (Clay et al., 1996). Other aspects related to ozone precursors will be discussed in section 3.3.*"

And:

"*Bogotá and Quito exposure levels are consistent with ozone profile in situ measurements over the South American tropics, where the 50th (5th) percentile was found to be less than 40 (10) nmol mol$^{-1}$ from the surface to 200 (700) hPa (Gaudel et al., 2024).*"

- Gaudel, A., Bourgeois, I., Li, M., Chang, K.-L., Ziemke, J., Sauvage, B., Stauffer, R. M., Thompson, A. M., Kollonige, D. E., Smith, N., Hubert, D., Keppens, A., Cuesta, J., Heue, K.-P., Veefkind, P., Aikin, K., Peischl, J., Thompson, C. R., Ryerson, T. B., Frost, G. J., McDonald, B. C., and Cooper, O. R.: Tropical tropospheric ozone distribution and trends from in situ and satellite data, EGUsphere [preprint], https://doi.org/10.5194/egusphere-2023-3095, 2024.

Figure S2: caption reads trend in ppb/yr while ppm/yr is shown in the figure.

**Answer**: Thank you for noticing this. However, following Copernicus's guideline for units, all ppbv and ppmv were changed to nmol mol$^{-1}$ and μmol mol$^{-1}$, respectively.

Chapter 3.3: for the interpretation of the $O_3$ change points along with the trends of the precursors. I wonder if you looked into the hourly data and the trends of the different percentiles there. At many location worldwide, it is often seen that the lowest values (percentiles) do show a positive trends (due to the reduction in NO and less $O_3$ titration) while the highest values show negative trends.

**Answer**: Thanks for raising these chemical processes. Although our methodology was not designed to evaluate hourly variability, we believed that it is important to include NO titration more explicitly in the text because it is directly related to ozone response in urban environments and facilitates the interpretation of results. We added at the beginning of Section 3.3: "... *Under this regime (high VOC-to-$NO_x$ ratio), less NO is available to titrate $O_3$ due to fewer vehicle emissions, and VOC oxidations initiated by hydroxyl radicals efficiently convert NO to $NO_2$, which photolysis produces $O_3$ (Monks et al 2015).*"

Based on our understanding of urban chemistry, from the **Figure** below, we can infer that the more dramatic drop in $NO_x$ (5th percentile) after the change point (in Santiago) produces a steeper increase in ozone (95th percentile). Similarly, a minor slope in the 95th percentile of $NO_x$ produces a minor slope in Ozone (5th percentile).

[Figure]

Following your comment about the interpretation of the $O_3$ change points, we would like to propose the $NO_2$ to $NO_x$ ratio (based on monthly means) to further explain the differences in photochemistry between cities and their chemical regimen changes. We added: "…*Thus, the CO-to-$NO_x$ ratio increase after the $NO_x$ change point provides a favorable scenario for efficient ozone formation. The latter observation is also evident in the $NO_2$ to $NO_x$ ratio, which typically decreases in winter due to higher NO primary emissions but increases in summer due to VOC oxidations. These oxidations involve reactions of hydroperoxyl ($HO_2$) and alkyl peroxy radicals ($RO_2$) with nitric oxide (NO), leading to the production of $NO_2$. Figure 8 illustrates the intense photochemical activity in Santiago, with notably higher $NO_2$ to $NO_x$ ratio (>0.6), particularly during the warm season (October to*

*March). In contrast, Bogotá exhibits lower NO₂ to NOₓ ratio, seldom exceeding 0.5, except during periods such as the COVID-19 lockdowns. This pattern suggests that NO₂ formation is typically suppressed under Bogotá's current chemical conditions."*

[Figure]

***Figure 8***. *Nitrogen dioxide to nitrogen oxides ratio based on surface monthly means. The orange dots indicate the first three months of every year for reference purposes. Change points with 95% confidence intervals are represented by a vertical red line (green line) for nitrogen oxides (carbon monoxide) and shaded red (green).*

Lines 299-300: "… These measurements have been accompanied by an increase in ozone since 2008." This refers to my comment just made above. You could doublecheck if you see that in all of your data, too. "… increase in ozone since 2018 …" Which metric are you referring to?

In the case of São Paulo we added:

*"From 2008 to 2020 ozone increased by about 4 nmol mol⁻¹ at the 50ᵗʰ percentile **(Table 3)**. Notably, many higher ozone anomalies occurred in the warmer months (Jan-Feb) and were more frequent after the last NOₓ change point detected in June 2013 ± [April 2011, August 2015], suggesting an intensification of photochemical activity. Consistently, **Figure 8** shows that the NO₂ to NOₓ ratio has increased significantly since the last NOₓ change point in 2013."*

Please also notice that **Figure 9** was modified according to the suggestions of Referee #2:

[Figure]

Lines 350-351. "We attributed these observed ozone trends to […] the establishment of volatile organic compound-limited regimes.". This looks like a firm statement that might require some more (model) analysis.

**Answer**: We agree with this comment. We reformulated: *"We attributed these observed ozone trends to a greater decrease in nitrogen oxides than in carbon monoxide, which resulted in chemical regimes that efficiently convert nitric oxide into nitrogen dioxide."*

Lines 363-364: "… the lack of quality control, which prevents the inclusion of additional existing measurements." Do you refer to the quality of the measurements here? As a group of South American scientists /experts in high-quality observations, do you have any suggestion to improve the situation? Training, workshops, development of common standards (if not available), development of common tools for quality control, …

**Answer**: Considering the existing infrastructure (investment), recognizing the technical expertise of the network operators, and the benefit of using measurement methods approved by the US EPA, the amount of data that does not meet the quality criteria in this research is, at least, controversial. In this regard, we are aware that many networks have different administrations, i.e., the implementation of procedures varies from city to city, even within the same nation. However, this aspect is not the only issue because the time series of particulate matter, the main air quality problem in many South American cities, is notoriously better in terms of data quality due, in part, to public pressure and prioritization made by environmental authorities. Therefore, to advance in this matter, it is fundamental to understand the importance of assigning the appropriate relevance to gaseous pollutants and make efforts towards homogenizing QA/QC procedures within the nation first and then in the entire South America.

We reword: *"…, while an improvement for future research requires elevating the quality assurance/quality control procedures to the same level, in each nation first, and then for South America. This action could mean a potential addition of up to 70 ozone time series (49%) rejected due to the controls we performed"*.

**Referee Comment (RC2)**

**General Comments:**

This paper reports on surface ozone trends in South America. Data from different locations are shown together with ozone precursor data, trends are reported and reasons for the trends are discussed with the help of the precursor data. Since publications of long time series of station data in South America are rare, the manuscript should be published after these questions have been answered:

**In line 152 the authors write:**

In these cities, a significant fraction of ozone precursors is emitted by vehicular fleets and has decreased according to air quality control measures such as the introduction of better fuel quality, sulfur content reduction, enforcement of threeway catalytic converters, stricter emission standards for new fleet vehicles and mandatory periodic technical inspection for inuse vehicles.

What is the impact of sulfur content reduction in fuel on ozone trends?

> **Answer**: We appreciate the comment. The sentence seems to indicate that sulfur reduction has a direct relationship to ozone formation when, in fact, sulfur reduction significantly reduced $SO_2$. For clarity, we have removed the mention of sulfur content.

**In Figure 3,** small boxes in the left panel could indicate where the zoomed regions in the right panel are.

> **Answer**: We modified all the figures and now we indicate where the cities are located.

**In Table 2**, the authors divide the MDA8 and peak season data into data for 2012-2016 and data for 2017-2021. The reason for this is not clear to me. Would it be possible to treat the MDA8 and peak season data like the datasets in Table 3 and calculate turning points, p and SNR values?

> **Answer**: The objective is to provide the community with the level of compliance of ozone metrics for intercomparison purposes on a time scale defined in the TOAR-II framework:
>
> https://igacproject.org/sites/default/files/2023-04/TOAR-II_Community_Special_Issue_Guidelines_202304.pdf
>
> To clarify this objective, we added in section 2.1:
>
> *"Present-day short- and long-term ozone exposure levels were assessed for all stations available following the TOAR-II recommendation for time scales, i.e., averaged values across 2017-2021, to facilitate intercomparisons with other studies."*
>
> And:
>
> *"Additionally, Table 2 lists the present-day ozone exposure metrics for the aggregated stations in the subdivisions used and includes the preceding five years (2012-2016) to provide a frame of comparison".*

Regarding the following question: Would it be possible to treat the MDA8 and peak season data like the datasets in Table 3 and calculate turning points, p and SNR values?

> **Answer**: Our primary research focus is to estimate the ozone's long-term trend, and to do that, we utilized monthly anomalies instead of other existing metrics. We did not determine the trend for exposure metrics, which provide one value per year according to WHO guidelines. Besides, we believe that providing the exposure status (presentday) makes it possible to evaluate the compliance of these exposure metrics. Also, notice that we are following the recommendations from the "Guidance note on best statistical for TOAR analyses" to make intercomparison across different studies easier, and many of them utilized monthly anomalies. Also, trends based on monthly anomalies *show the influence of unforced climate variability on interannual ozone fluctuations* (Cooper et al., 2020). In addition, monthly anomalies produce trend estimates with less uncertainty (when dealing with missing data) than other metrics with higher variability (daily or monthly averages).

- Cooper, O. R., Schultz, M. G., Schröder, S., Chang, K. L., Gaudel, A., Benítez, G. C., Cuevas, E., Fröhlich, M., Galbally, I. E., Molloy, S., Kubistin, D., Lu, X., McClure-Begley, A., Nédélec, P., O'Brien, J., Oltmans, S. J., Petropavlovskikh, I., Ries, L., Senik, I., Sjöberg, K., Solberg, S., Spain, G. T., Spangl, W., Steinbacher, M., Tarasick, D., Thouret, V., and Xu, X.: Multi-decadal surface ozone trends at globally distributed remote locations, Elementa, 8, https://doi.org/10.1525/elementa.420, 2020.

In **line 210** the authors write: Regardless of the latitude of each large city analyzed, each urban agglomeration contains subdivisions with high-certainty positive ozone trends.

This is difficult to see from the data in Table 3: in the Bogotá region, the trend is either positive or negative, depending on whether you look at the 5th, 50th or 95th percentile. Also, this is in contradiction to what the authors write in line 243: The ozone mixing ratios in Bogotá showed no evidence of reduction or increase during the last decade despite efforts to reduce primary pollutant emissions, as shown in Figure 4a.

I would suggest that the author rephrase the sentence or indicate which data set the authors are referring to.

> **Answer**: We rephrase as follows: "Santiago and São Paulo exhibited clear positive trends with high or very high certainty after the change points in 5th, 50$^{th}$, *and 90$^{th}$ percentiles (Table 3)*".

**Table 3** is central to the manuscript. For better readability, I would suggest including a column where it is easy to see whether a trend is certain according to the criteria in Table 1, e.g. very high certainty, high certainty…

> **Answer**: We appreciate the suggestion that was implemented in **Table 3**.

In **line 224** the authors write: the trend observed after 2014 was likely impacted by the COVID-19 pandemic in 2020 and possibly in 2021 (Putero et al., 2023).

This would suggest a change point in 2019/2020.

Can this be seen in the data?

> **Answer**: Please notice that according to our methodology, "*We imposed a minimum period of 4 years after the occurrence of a change point to avoid detection at the extremes of the time series*", so, we are not considering those possible change points. However, as shown in **Table 3**, it is worth noting that the period (2014-2021) presents low or very low certainty (5th, 50th, and 90th). To make this point clearer, we added:

> *"On the other hand, the second change point confidence interval (2014) was very wide compared with the length of the time series, as shown in **Figure 4**, and although the trend for this period was still positive (0.07 nmol mol$^{-1}$ year$^{-1}$), the certainty was relatively low. Moreover, the trend observed after 2014 was likely impacted by ozone*

*drops due to the COVID-19 pandemic in 2020 and possibly in 2021 as found at high-elevation sites, mainly in the northern hemisphere (Putero et al., 2023)."*

In **line 239** the authors write: In general terms, we note that these ozone precursor abatement measures have been implemented, ignoring the VOC-to- $NO_x$ ratio, suggesting that ozone increases once the VOC-limited regime is reached. The latter, together with the extensive wildfires around the cities studied, could explain the occurrence of trend change points at some sites. This should be discussed more in detail.

The authors write that ozone trend is determined by ozone production. What is the role of titration effects, e.g. an increase in ozone concentration due to a decrease in NO mixing ratio?

> **Answer**: We agree that an explicit mention of the $NO_x$ role would be welcome. We added: "…*Under this regime (high VOC-to-$NO_x$ ratio), less NO is available to titrate $O_3$ due to fewer vehicle emissions, and VOC oxidations initiated by hydroxyl radicals efficiently convert NO to $NO_2$, which photolysis produces $O_3$ (Monks et al 2015). Moreover, biogenic VOCs are known as effective ozone precursors due to their high reactivity toward OH (Atkinson and Arey, 2003). Unfortunately, in South America, biogenic VOC measurements are derived from field campaigns conducted for short time periods; therefore, the direct influence of biogenic VOCs on ozone levels is beyond the scope of this study. However, since higher vegetation releases are expected to occur during months impacted by extreme temperatures and heat waves, their indirect effect on the summer months is likely to be reflected in ozone trends.*"

The CO trend reflects the trend of VOCs originating from combustion. What is the role of biogenic VOCs in ozone formation here?

> **Answer**: Please refer to the previous answer.

A change in $NO_x$ or VOC concentration would have an opposite effect on ozone production, depending on whether the chemical regime is VOC-limited or $NO_x$-limited. Is there any indication of whether ozone production is VOC-limited or $NO_x$-limited?

> *Answer*: In the introduction, we provided a frame and references supporting this claim*: "In most urban areas with adequate monitoring coverage allowing characterization of the temporal and spatial variabilities in ground-level ozone, a chemical regime of ozone formation limited by volatile organic compounds (VOCs) has been found in previous work (Elshorbany et al., 2009; Seguel et al., 2020; Silva et al., 2018; Silva Júnior et al., 2009). This chemical regime was also observed during COVID-19 pandemic lockdowns, when several cities experienced increased ambient ozone mixing ratios (e.g., Bogotá, Quito, Santiago, São Paulo and Lima) due to a decrease in nitric oxide (NO) emitted by motorized transportation vehicles (Seguel et al., 2022; Sokhi et al., 2021; Cazorla et al., 2021b)."*

If wildfires play a role. I would expect different trends in different seasons? Has this been studied?

> **Answer**: We believe that mega-fires are playing a role in central Chile, as shown in Figure B1 and as indicated by the anomaly of February 2023 (**Fig. 7**). We did not explore seasonal trends in this research because our approach was more oriented to determining to what extent fires affect or impact the trends and also to look at specific fire events for process understanding.

In **line 244** the authors write: However, in the northern area of the city, which is impacted by ozone formation in higher proportions, the median ozone trend decreased at a rate of -1.01ppb yr-1 (high certainty) between 2008 and 2013.

Can it be shown that the northern area is more affected by ozone precursors? I would expect the ozone trend of the whole region to be shown and compared with the ozone trend of the northern region. The same comparison should be made for precursors.

Answer: We appreciate this comment because, together with another suggestion, requesting the uncertainty of the change point (below) is crucial to conveying our message better at this point. We believe the higher exposure to ozone can be seen in **Table 2**, where differences between south and northern (and central) Bogotá are well depicted. However, our main point here is to highlight that despite the decrease in CO (before the change point) and NOx, the ozone remains almost steady. Therefore, we prefer to eliminate the mention of the northern part of Bogotá. We reworded: *"Panels c and d in **Figure 4** also show the trends in CO and $NO_x$ and their change point with 95% confidence interval to illustrate the low ozone sensitivity under this chemical regime, i.e., before and after the precursors' change point."*

In **line 264** the authors write: Overall, in the Quito $NO_x$-saturated environment, decreases in $NO_x$ precursors were anticorrelated with increases in ozone.

This sentence is unclear to me as it is not clear what $NO_x$ saturation means. Is ozone production limited by VOC? Does it mean that $O_3$ is removed by titration with NO? Also, the term $NO_x$ precursors is not clear. Are the authors referring to the sources of $NO_x$, e.g. that $NO_x$ emissions have decreased?

**Answer**: We appreciate the observations. We reframe as follows:

*"In turn, the relationship between $NO_2$ and CO with ozone is more evident before the precursors' change point when the increase in the $NO_2$ anticorrelates with ozone. This behavior allows us to infer higher ozone destruction due to NO titration (note that NO is not measured in Quito)."*

In **line 264** the authors write: decreases in $NO_x$ precursors were anticorrelated with increases in ozone.

Shouldn´t it be: NOx mixing ratio was anticorrelated to the ozone mixing ratio.

**Answer**: Please refer to the previous answer

In **line 267** the authors write: This change generally coincided with the time series period when the ozone trend stopped decreasing, leading to a change point.

However, the change point was for ozone was in 2011, whereas the change points for $NO_x$ and CO were in 2013 and 2014. Is this within the uncertainty of the change point determination?

**Answer**: We much appreciate this comment. We added the change point confidence interval to all Figures. In the case of Quito, we can see now how the change points overlap particularly for CO and $NO_2$ between Aug 2012 and Dec 2013.

**Figure 6:**

[Figure]

In **line 267** the authors write: The implementation of this policy probably shifted the composition and proportion of precursors, especially during the morning.

Can this be shown? Here I would expect that the trend should be more significant when looking only at the values during the morning hours at polluted sites.

> **Answer**: As discussed above, our methodology was not designed to analyze such hourly time variations. Therefore, for consistency, we eliminate the mention of the morning hour restriction. We reworded: "*The change points of CO and NO₂ detected in April 2013 ± [August 2012, December 2013] and July 2013 ± [May 2012, September 2014] overlapped with ozone change point, indicating a shift from higher ozone inhibition observed around the precursors' change point to a more uncertain regime (Figure 6). Also, the reliability of the CO and NO₂ trends is low or very low certainty after 2013.(**Figure 6a**).*"

In **line 275** the authors write: Ozone in Santiago decreased for nearly two decades due to public policies focusing mainly on curbing particulate matter.

This implies that PM and $O_3$ have the same sources. Can this be shown? Do PM and $O_3$ have the same trend?

> **Answer**: To avoid any ambiguity in the meaning of the sentence, we reword: "*Ozone in Santiago decreased for nearly two decades due to public policies focusing mainly on curbing particulate matter in cold seasons. Thus, particles have decreased over time (Jorquera et al., 2020), while the benefits of those policies turned ineffective in curbing ozone during warm seasons, particularly in the last decade (Seguel et al., 2020).*".

In **line 282** the authors write: In other words, until 2017, the policies effectively lowered the highest ozone percentiles.

From the plots it looks that ozone was low because $NO_x$ was high and NO reacted with ozone to convert ozone into $NO_2$. The question is what happened in 2017? Did the policy change or was $NO_x$ so low that less ozone was removed by titration?

> **Answer**: We agree with this comment. Santiago is characterized by a high rate of conversion of NO to $NO_2$ which can be studied through the $NO_2$ to $NO_x$ ratio:

[Figure]

To the best of our knowledge, the change point in 2017 is explained by several processes that intensified in the last decade, some of them addressed in **Section 3.4**. However, our understanding of urban chemistry is that NO is more depleted during heat waves and biogenic releases (isoprene) increase. Also, during fires, ozone is transported to Santiago, as shown in Figure B1. Regarding policies mainly focused on transportation achieved reductions of direct emissions, but at the same time Santiago Sprawled, including motor vehicles, in the last decade counteracting the measures.

In **line 303** the authors write: Notably, many higher anomalies occurred in the warmer months (Jan-Feb) and were more frequent after the ozone change point in 2008 (Figure 7a).

This is not readily apparent from the figure. The orange dots indicating the beginning of the year are on either side of the trend line. Perhaps it is possible to specify where the trend manifests itself.

**Answer**: We added the change point uncertainty in **Figure 9** to facilitate the interpretation and we also reworded: *"Figure 8 shows that CO decreased throughout the period, with a change point in September 2008 ± [September 2006, September 2010], which overlapped with the ozone change point detected in March 2008 ± [September 2006, September 2009]. From 2008 to 2020 ozone increased by about 4 nmol mol$^{-1}$ at the 50$^{th}$ percentile (**Table 3**). Notably, many higher ozone anomalies occurred in the warmer months (Jan-Feb) and were more frequent after the last NO$_x$ change point detected in June 2013 ± [April 2011, August 2015], suggesting an intensification of photochemical activity. Consistently, **Figure 8** shows that the NO$_2$ to NO$_x$ ratio has increased significantly since the last NO$_x$ change point in 2013."*

**Figure 9**:

[Figure]

In **line 304** the authors write: As a result, the ozone trends at the 90th and 95th percentiles increased

I would suggest writing: accordingly or correspondingly (as this is not a result, but a different way of presenting the results)

**Answer**: We modify the text as suggested

In **line 326** the authors write: In this regard, extreme positive ozone anomalies were observed in January 2017 (7.6 ppb) and February 2023 (8.6 ppb), caused by ozone and precursors transported from areas affected by intense wildfires (Fig 6a).

Can these datapoints be shown in the figure?

**Answer**: We modified **Figure 7** to show those anomalies.

In **line 342** the authors write: Short-term (MDA8) and long-term (peak-season) exposure metrics calculated for the present day (2017-2021) revealed latitudinal differences in South America.

However, in lines 343 and following, the authors argue that there are several factors, and that latitude plays only a minor role. I would suggest omitting the word "latitudinal" here.

**Answer**: Following the suggestion, we omitted the word and rephrase as follow: "*Short-term (MDA8) and long-term (peak-season) exposure metrics calculated for the present day (2017-2021) revealed differences between the tropics and extratropics in South America*".

In **line 350** the authors write: We attributed these observed ozone trends to a greater decrease in nitrogen oxides than in carbon monoxide, which resulted in the establishment of volatile organic compound-limited regimes.

The arguments in favour of this statement, as set out in line 239, are in my opinion too weak, see comments on line 239.

**Answer**: We agree with this observation, so accordingly we eliminated the second part of the sentence. We reformulated: "*We attributed these observed ozone trends to a*

*greater decrease in nitrogen oxides than in carbon monoxide, which resulted in chemical regimes that efficiently convert nitric oxide into nitrogen dioxide."*

In **line 366** the authors write: Finally, our results revealed signs of a climate penalty for ozone in South America

How would this climate penalty be reflected in the data? Is this related to the more frequent forest fires? This could then be mentioned again here.

> **Answer**: We rephrase: *"Finally, our results revealed signs of a climate penalty for ozone in South America, derived from more favorable meteorological conditions for wildfire propagation in Chile and extensive heat waves in southern Brazil. In these regions' urban environments, the increase in ozone poses the highest risk."*

In **line 366** the authors write: and identified extratropical zones as those where the increase in ozone poses the highest risk.

As the authors also argue, latitude is only one of several factors that determine the ozone trend, in addition to local measurements as in Quito, Santiago, and São Paulo or the local wind system as in Bogota, see also line 150 ff. A dependence of the ozone trend on the latitude would have to be shown specifically on the existing data.

> **Answer**: Please see the previous answer.

**Community Comments (CC) by Owen R. Cooper**

Major Comments:

The authors have provided the first continent-wide overview of surface ozone across South America, based on all available observations. This is an important topic and a welcome addition to the TOAR-II Community Special Issue. The trend analysis is extremely well done, and the authors have done a very good job of following the recommendations from the "Guidance note on best statistical for TOAR analyses". In particular the use of the TOAR vector approach for visualizing trends and the use of the TOAR color table makes it very easy to compare these new TOAR results to previous TOAR studies.

Overall the findings are consistent with the papers from TOAR-I and with the papers submitted so far to the TOAR-II Community Special Issue. A paper that is currently under review with the TOAR-II Community Special Issue reports boundary layer and free tropospheric ozone observations across the tropics based on IAGOS commercial aircraft and ozonesondes. It would be helpful if the authors could briefly discuss how their findings are relevant to these other new results, or how the ozone values in the free troposphere (reported by Gaudel et al., 2024) might affect the surface sites:

Gaudel, A., Bourgeois, I., Li, M., Chang, K.-L., Ziemke, J., Sauvage, B., Stauffer, R. M., Thompson, A. M., Kollonige, D. E., Smith, N., Hubert, D., Keppens, A., Cuesta, J., Heue, K.-P., Veefkind, P., Aikin, K., Peischl, J., Thompson, C. R., Ryerson, T. B., Frost, G. J., McDonald, B. C., and Cooper, O. R.: Tropical tropospheric ozone distribution and trends from in situ and satellite data, EGUsphere [preprint], https://doi.org/10.5194/egusphere-2023-3095, 2024.

> **Answer**: We appreciate this suggestion: We added in **Section 3.1**: *"Bogotá and Quito exposure levels are consistent with ozone profile in situ measurements over the South American tropics, where the $50^{th}$ ($5^{th}$) percentile was found to be less than 40 (10) nmol mol$^{-1}$ from the surface to 200 (700) hPa (Gaudel et al., 2024)."*

Mixing ratios are reported in units of ppbv, however, Copernicus journals require units of nmol mol$^{-1}$.

**Answer**: We changed the units from ppbv (ppmv) to nmol mol$^{-1}$ (μmol mol$^{-1}$) throughout the document.

Line 358

Tololo is mentioned in the Conclusions and acknowledged as being a valuable monitoring station, but it is not mentioned much in the main text. A figure showing the full Tololo time series, along with its change points (e.g. Figure 7a), would be very helpful and would clearly illustrate the shifts in background ozone.

**Answer**: We added some additional lines to the original text: *"At Tololo, an upward ozone trend of 0.29 nmol mol$^{-1}$ year$^{-1}$ was observed between 2006 and 2014 (**Fig. 4**), with a very high certainty (**Table 3**). Within this period (2006-2004), the higher percentiles (> 50$^{th}$) displayed the most significant increasing trends (> 0.38 nmol mol$^{-1}$ yr$^{-1}$). Notably, the trend change point in May 2006 ± [November 2002, November 2009] coincides with the global methane increase after the plateau observed between 1999 and 2006 (Lan et al., 2024). These observational findings could be explored further by comparing them with outputs from regional models capable of quantifying the ozone increase associated with methane changes. On the other hand, the second change point confidence interval (2014) was very wide compared with the length of the time series, as shown in **Figure 4**, and although the trend for this period was still positive (0.07 nmol mol$^{-1}$ year$^{-1}$), the certainty was relatively low. Moreover, the trend observed after 2014 was likely impacted by ozone drops due to the COVID-19 pandemic in 2020 and possibly in 2021 as found at high-elevation sites, mainly in the northern hemisphere (Putero et al., 2023)."*

And we added the figure suggested (**Fig. 4**):

[Figure]

**Figure 4**: *Percentile trends derived by quantile regression based on the monthly surface ozone (Panel a) in Tololo. The orange dots in Panel a, indicate the first three months of every year for reference purposes. In Panel a, the red line corresponds to the 50$^{th}$ percentile, and the light blue lines correspond to the remaining percentiles. Change points with 95% confidence intervals are represented by a vertical red line and shaded red (first change point) and blue light (second change point).*

Data Availability statement:

Please provide additional details that will allow the reader to find the data. 1) provide a link to the TOAR-II surface ozone database

2) provide a link to the MODIS data

3) Figure B1 shows observations of CO and NOx. Are these data available from the TOAR database?

4) to acknowledge the WMO GAW program, please also provide a link to the location where GAW data can be downloaded: https://ebas.nilu.no/

> **Answer**: We added in the section of Data Availability: *"Datasets used in this research are available in the TOAR-II database: https://toar-data.org. Active fires can be downloaded at: https://modis-fire.umd.edu/. Carbon monoxide and nitrogen oxides surface data (Fig. B1) can be downloaded at: https://sinca.mma.gob.cl. GAW data can be downloaded at: https://ebas.nilu.no/."*

**Minor Comments:**

Abstract: The way the first sentence is written, the subject of the sentence is "trends" and not "ozone". Therefore, the word "precursors" refers to trends, and not ozone. A better way to phrase the sentence is:

*"In this study, trends of 21st-century ground-level ozone and ozone precursors were examined across South America, an understudied region where trend estimates have rarely been comprehensively addressed."*

> **Answer**: We are very grateful for the suggested improvement which has been incorporated into the main text.

Introduction, first paragraph:

Section 2.2.5.3 in Chapter 2 of IPCC AR6 (Gulev et al., 2021) provides a concise summary of global tropospheric ozone trends, based on the TOAR findings. I recommend that the IPCC findings be used as the starting point for the trend discussion in the submitted manuscript:

"Since the mid-1990s, free tropospheric ozone has increased by 2–7% per decade in most regions of the northern mid-latitudes, and 2–12% per decade in the sampled regions of the northern and southern tropics (high confidence). Limited coverage by surface observations precludes identification of zonal trends in the SH, while observations of tropospheric column ozone indicate increases of less than 5% per decade at southern mid-latitudes (medium confidence)."

Gulev, S.K., P.W. Thorne, J. Ahn, F.J. Dentener, C.M. Domingues, S. Gerland, D. Gong, D.S. Kaufman, H.C. Nnamchi, J. Quaas, J.A. Rivera, S. Sathyendranath, S.L. Smith, B. Trewin, K. von Schuckmann, and R.S. Vose, 2021: Changing State of the Climate System. In Climate Change 2021: The Physical Science Basis.

Contribution of Working Group I to the Sixth Assessment Report of the Intergovernmental Panel on Climate Change [Masson-Delmotte, V., P. Zhai, A. Pirani, S.L. Connors, C. Péan, S. Berger, N. Caud, Y. Chen, L. Goldfarb, M.I. Gomis, M. Huang, K. Leitzell, E. Lonnoy, J.B.R. Matthews, T.K. Maycock, T. Waterfield, O. Yelekçi, R. Yu, and B. Zhou (eds.)]. Cambridge University Press, Cambridge, United Kingdom and New York, NY, USA, pp. 287–422, doi:10.1017/9781009157896.004

**Answer**: We modified the first paragraph considering the summary provided in the Section 2.2.5.3 (AR6, IPCC):

*"The global tropospheric ozone (O$_3$) burden has increased by 45% (109 ± 25 Tg) with medium confidence from 1850 to the present day due to anthropogenic precursor emissions (Szopa et al., 2021). Additionally, surface ozone has increased by 32-71% with large uncertainty in rural air across the Northern Hemisphere relative to historical observations (1896-1975) (Tarasick et al., 2019). Since the mid-1990s, free tropospheric ozone has increased with high confidence by 1-4 nmol mol$^{-1}$ decade$^{-1}$ in most regions across the northern mid-latitudes and 1-5 nmol mol$^{-1}$ decade$^{-1}$ within the tropics (Guleb et al., 2021). In contrast, the identification of ozone trends in the Southern Hemisphere, including South America, is precluded due to the limited coverage by ground-level monitoring stations, while observations of tropospheric column ozone since the mid-1990s indicate* medium confidence *increases of less than 1 nmol mol$^{-1}$ decade$^{-1}$ at southern mid-latitudes (Gulev et al., 2021, Cooper at al., 2020)."*

Line 48

When summarizing ozone standards, the averaging time should also be given. For example, do the values 51-71 ppbv refer to the maximum daily 8-hour average?

**Answer**: We modified the text accordiggly: "*Most countries set ozone standards based upon an 8-h average, ranging from 51 to 71 nmol mol$^{-1}$*"

Line 59

Please specify that GAW is a WMO program.

**Answer**: We added the specification: "*Monitoring has also been implemented through the Global Atmospheric Watch (GAW) program of the World Meteorological Organization (WMO) at remote locations*"

Line 104

The WHO AQ guidelines report is missing from the list of references: WHO global air quality guidelines. Particulate matter (PM2.5 and PM10), ozone, nitrogen dioxide, sulfur dioxide and carbon monoxide. Geneva: World Health Organization; 2021. Licence: CC BY-NC-SA 3.0 IGO.

**Answer**: We added the reference.

Line 220

Ozone increased at Tololo from 2006 to 2014 by about 2.3 ppbv. The discussion seems to imply that this increase of ozone is due to the increase of methane over the period 2006-2014. The observed methane increase from 2006 to 2014 (according to NOAA GML: https://gml.noaa.gov/ccgg/trends_ch4/) was about 50 ppbv, or about 3%. While methane drives the background increase in ozone, are there any modelling results that can support this suggestion? A recent submission to the TOAR-II Community Special Issue (Nalam et al., 2024) calculates the change in global surface ozone due to methane increases over the period 2000-2018. For the period 2006-2014, Figure 7b, indicates an ozone increase of no more than 1 ppbv.

Nalam, A., Lupascu, A., Ansari, T., and Butler, T.: Regional and sectoral contributions of NOx and reactive carbon emission sources to global trends in tropospheric ozone during the 2000–2018 period, EGUsphere [preprint], https://doi.org/10.5194/egusphere-2024-432, 2024.

**Answer**: This is an excellent comment, and we thank the reference. Since this is an observational-based study, we did not apply regional models. However, we believe this may be feasible to address in the ongoing TOAR-II Regional Assessment of Tropospheric Ozone over South America.

We added in the text: "*These observational findings could be explored further by comparing them with outputs from regional models capable of quantifying the ozone increase associated with methane changes.*"

Table 1

Please check the greater-than-or-equal-to symbols in this table against the original Table 3 in the "Guidance note on best statistical for TOAR analyses". Many of these symbols don't match the original table.

**Answer**: We double-checked and made the changes according to the original Table.

For those of us not highly familiar with the geography of Chile, it's not clear that the cluster of observing stations near the top of the map is located in the Santiago urban area. Can this be indicated on the map?

**Answer**: We made the suggested changes.